



# CFD modeling of actual eroded wind turbine blade

Kisorthman Vimalakanthan, Harald van der Mijle Meijer, Iana Bakhmet, and Gerard Schepers

TNO, Westerduinweg 3, 1755 LE Petten, Netherlands

**Correspondence:** Kishore (k.vimalakanthan@tno.nl)

**Abstract.** Leading edge erosion (LEE) is one of the most critical degradation mechanisms that occur with wind turbine blades (WTBs), generally starting from the tip section of the blade. A detailed understanding of the LEE process and the impact on aerodynamic performance due to the damaged leading edge (LE) is required to select the most appropriate Leading Edge Protection (LEP) system and optimize blade maintenance. Providing accurate modeling tools is therefore essential.

This paper presents a two-part study investigating Computational Fluid Dynamics (CFD) modeling approaches for different orders of magnitudes in erosion damage. The first part details the flow transition modeling for eroded surfaces with roughness in the order of 0.1-0.2mm, while the second part focuses on a novel study modeling high-resolution scanned LE surfaces from an actual blade with LEE damage in the order of 10-20mm (approx. 1% chord). 2D and 3D surface resolved Reynolds Average Navier Stokes (RANS) CFD models have been applied to investigate wind turbine blade section in the Reynolds number range
of 3-6million.

From the first part, the calibrated CFD model for modeling flow transition accounting roughness shows good agreement of the aerodynamic forces for airfoils with leading-edge roughness heights in the order of 140-200um, while showing poor agreement for smaller roughness heights in the order of 100um. Results from the second part of the study indicate that up to 3.3% reduction in AEP can be expected when the LE shape is degraded by 0.8% of the chord, based on the NREL 5MW
turbine. The results also suggest that under fully turbulent condition the eroded LE shapes show the least amount of influence on the aerodynamic performances and results in negligible difference to AEP.

## 1   Introduction

Leading edge erosion (LEE) is one of the most critical degradation mechanisms that occur with wind turbine blades (WTBs), generally starting from the tip section of the blade. The tip section, due to the higher speeds, contributes to the majority of the
torque production. The last 15% of the blade radius for a multi-megawatt rotor can contribute up to 25% of the total power production and more than 5% reduction of annual energy production (AEP) for a wind farm. Eroded blades result in sub-optimal rotor performance and a significant loss of AEP of the wind turbine. In addition, the influence of surface roughness in the form of erosion or contamination is of great practical importance for many flow applications and particular interest to the wind industry. This form of the blade degradation process is known to accelerate the boundary layer transition to result in
substantial loss of the blade's aerodynamic performance (Sareen et al., 2014).



The higher flow speeds at the blade tip also result in a higher impact velocity of the rain droplets on the blade surface which shortens the incubation time and increases the erosion damage rate on the leading edge (LE) of the blade. A detailed understanding of the LEE process and the impact on aerodynamic performance due to the damaged LE is required to select the most appropriate Leading Edge Protection (LEP) system and optimize blade maintenance. Providing accurate modeling tools is therefore essential. This paper presents a two-part investigation, where the first part focuses on accurate prediction tools for simulating various roughness distributions (in the order of 0.1-0.2mm) due to light erosion or contamination and its influence on boundary layer transition. The second part will focus on explicit modeling high resolution scanned LE surfaces from an actual blade with LEE damage in the order of 10-20mm.

The novel aspect of this work is the modeling of high-resolution LE surfaces of an actual blade with LEE damage, that was captured in the field using state-of-art optical 3D scanning technologies. This makes this study different from numerically assumed damaged blades with standardized damage profiles on the LE. This method is applicable for an indication of the effect of actual damage on the performance. Secondly, the scanning method offers the possibility to monitor damage to the blade with inspections, which has added value for an operator in choosing the right O&M and control strategy for the wind turbine.

Using the point cloud data, 2D and 3D Reynolds Average Navier Stokes (RANS) Computational Fluid Dynamics (CFD) models have been applied to investigate the influence of shape change at the LE on aerodynamic performance. Both models are shown to predict the change in flow characteristics due to the LE differences. The simulations indicated that depending on erosion conditions the lift and drag coefficients can be reduced and increased, respectively. The results from CFD simulation have been used in a Blade Element Momentum (BEM) model to calculate and establish the performance of the WTBs in terms of power coefficient and relative changes in AEP for both clean and eroded cases.

## 2    CFD transition model for rough surfaces (Part 1)

The influence of surface roughness in the form of erosion or contamination is of great practical importance for many flow applications and particular interest to the wind industry. This form of the blade degradation process is known to accelerate the boundary layer transition to result in substantial loss of the blade's aerodynamic performance (Sareen et al., 2014).

The boundary layer transition process is an extremely complicated process that has been studied extensively for almost a century. During this phenomenon, the flow characteristics are changed from a streamlined laminar flow to a chaotic turbulent flow. Modeling flow transition for wind turbine blades has previously shown promising results for validating computational models (Vimalakanthan, 2014). Accurate prediction tools for simulating various roughness distributions and their influence on boundary layer transition are therefore fundamental for efficient design optimization and implementation of wind power systems.

The objective of this part of the study was to develop and calibrate the roughness amplification model for the open source flow solver OpenFOAM (Jasak et al., 2007). This model was originally envisaged by Dassler, Kozulovic, and Fiala (Dassler et al., 2010, 2012) in 2010, and recently in 2017 Langel et al from Sandia National Laboratory published a detailed thesis



(Langel et al., 2017b) on this model implementation into the flow solver OVERFLOW-2. Langel et al work was used as the basis for this development and calibration of the roughness amplification model for OpenFOAM.

This model implementation allows for 2D and 3D boundary layer transitional CFD simulations including the effect of surface roughness. This is achieved via an additional transport equation for modeling roughness onto a pre-existing transition model. This approach uses an additional field variable, the roughness amplification quantity ($Ar$) to be transported downstream to generate a region of influence due to the prescribed roughness, thus the underlying transition model is triggered accounting for the effect of surface roughness.

Firstly, the results from validating the underlying transition model within the OpenFOAM solver are applied to clean surfaces. For verification purposes, the results are also compared against the panel code XFOIL. Then the implementation of the transport equation based roughness model and its calibration study for airfoil sections with rough leading edge are presented accordingly.

The work was funded under the AIRTuB project (Automatic Inspection and Repair of Turbine Blades) with a Top Sector
Energy subsidy from the Ministry of Economic Affairs of the Netherlands.

## 2.1 Simulation setup - OpenFOAM

The steady-state incompressible simpleFoam solver was used for all OpenFOAM simulations. The name comes from the fact that simpleFoam uses the Semi-Implicit Method for Pressure-Linked Equations (SIMPLE) algorithm of Patankar and Spalding to enforce the pressure/velocity coupling. It is a steady-state solver for incompressible turbulent flows. It can be used with a
variety of turbulence models which are available in OpenFOAM. The momentum equations are solved using the 2nd order bounded Gauss linearUpwind scheme. The turbulence closure relations are solved using the bounded Gauss limitedLinear numerical scheme that limits towards 1st order upwind in regions of rapidly changing gradient, while other parts of the domain are resolved using 2nd order linear scheme to achieve greater numerical stability.

Following OpenFOAM solvers were used for solving the corresponding equations describing the system.

– The momentum, k, omega, ReTheta, gamma and Ar equations:

The smoothSolver is used with a symmetric GaussSeidel smoother.

– The pressure equation:

The GAMG (Geometric-algebraic multi-grid) solver was used for pressure equation. GAMG solver also requires a smoother for its operation, thus once again GaussSeidel smoother was used with GAMG to ensure a faster convergence.
The size of the initial coarse mesh is specified through in the nCellsInCoarsestLevel entry, which was specified to be 1000 based on 24cpu computations. The agglomeration of cells is performed by the selected FaceAreaPair method.

Numerical convergence was assessed based on the RMS residual for all equations < 1E-7 for all simulations with a steady hysteresis on the lift and drag forces (Figure 1).

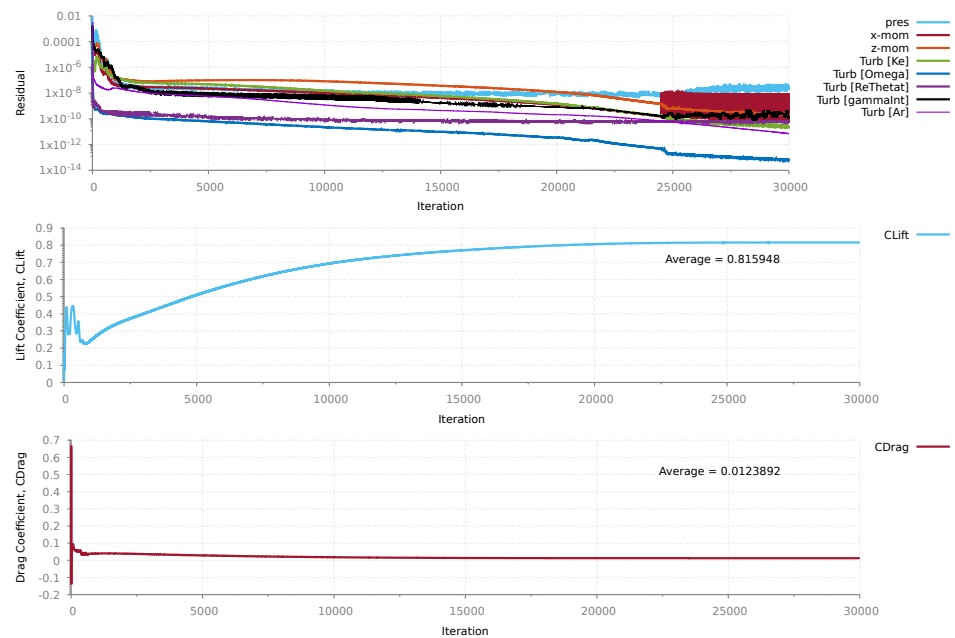

**Figure 1.** Computed residuals for NACA 63-418 with rough leading edge with 200um roughness height at AoA=4°

## 2.2 Flow transition prediction on clean surfaces - Validation/Verification study

The roughness amplification model uses Langtry-Menter's local correlation based transition model (SSTLM) (Langtry, 2006; Langtry and Menter, 2009) as the underlying model to trigger boundary layer transition, while the fully turbulent parts of the domain are resolved with Menter's k-Omega SST model (Menter, 1992). In order to establish the performance of the underlying transition model, two test cases were studied to verify and validate with experimental measurements.

The first test case was conducted using the EROCOFTAC T3A experiments studying a flat plate geometry (Savill, 1993), 95 which has become a renowned benchmark case for validating computational transition models. The simulated boundary conditions are presented in Figure 2. Due to the inherent decay of turbulence with RANS modeling, a slightly higher turbulence intensity (3.7%) compared to the experimented 3% was prescribed at the inlet to match the value measured at the start of the flat plate. Based on the results (Figure 3) the underlying transition model (SSTLM) accurately predicts the decay of measured turbulence intensity, however, the transition point is predicted earlier than measured.

The second test case was conducted using Sandia's experimental study (LEES-Dataset) (David et al., 2020) using NACA 63-418 airfoil at three different Reynolds numbers (Re= 2.4E6, 3.2E6 and 4.0E6). RANS 2D CFD simulations were performed with a domain extent of 90 chords was used to model the far-field (Figure 4). The results from the grid refinement study (Figure 5) showed that a minimum of 350points (Mesh: Medium) are required to resolve the airfoil section to achieve a grid-independent solution. Similarly, a study assessing different initial grid heights normal to the wall (Figure 6) has reviled that a



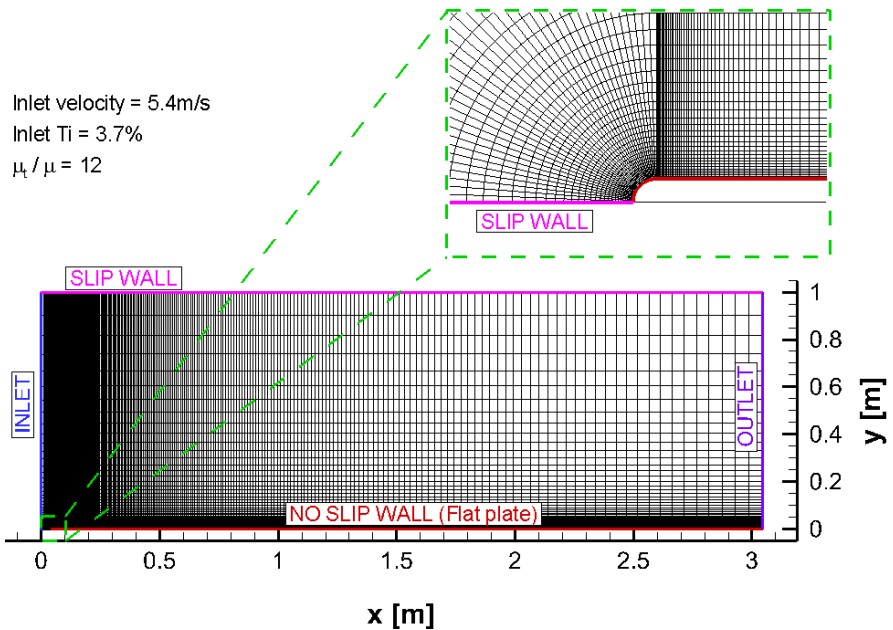

**Figure 2.** Turbulence kinetic energy and Wall shear stress along clean flat plate

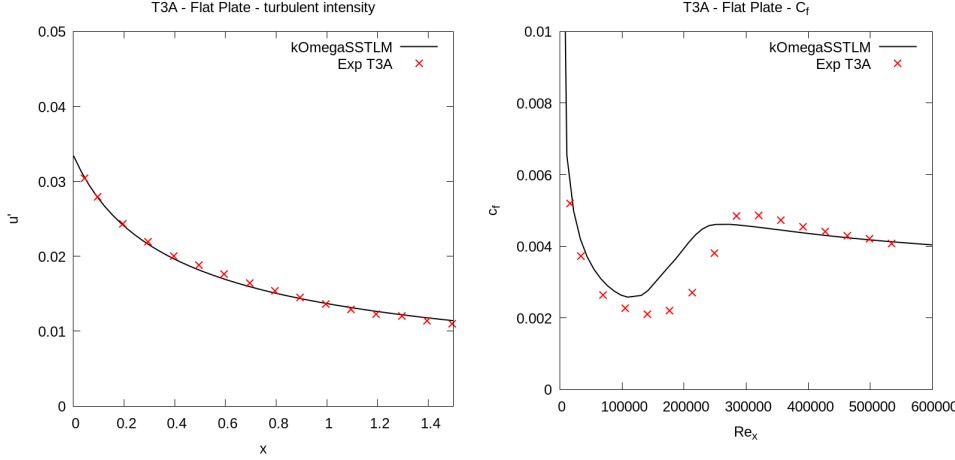

**Figure 3.** Turbulence kinetic energy and Wall shear stress along clean flat plate

minimum y+ value of 1 is required to achieve physical transitional results. The transition location was evaluated based on the rapid increase in intermittency variable (gamma), as detailed in (Vimalakanthan, 2014).

In the initial stages of this investigation, the original NACA 63-418 airfoil geometry from Abbott & von Doenhoff(Abbott et al., 1945) was used to verify numerical accuracy as also used for validation study by Wilcox (Wilcox et al., 2017). However,





there has been some uncertainty about the exact geometry used in the experiment. Different publications using this experimental

dataset(LEES-Dataset) for validating their numerical models have considered slightly different geometries, such as using the original geometry (Wilcox et al., 2017) defined by Abbot & von Doenhoff (Abbott et al., 1945), while others have considered the geometric definition using a Bezier curve (Langel et al., 2017a).

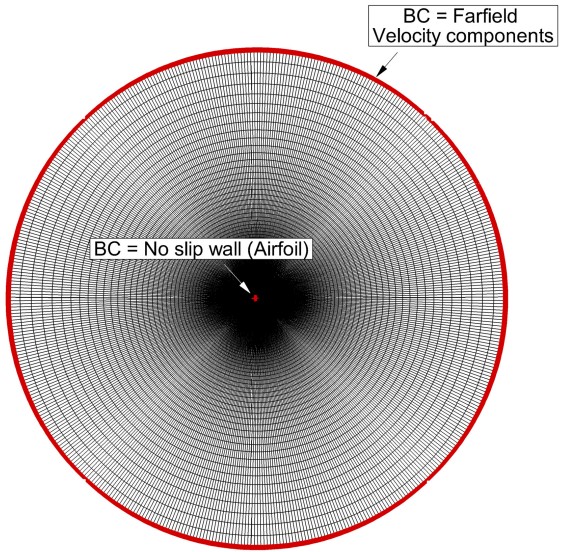

**Figure 4.** O-grid with 90chord domain extent showing the Full Hex grid with boundary conditions

The reason which pleads for the Bezier geometry is that in their publication (Langel et al., 2017a) Langel et al have provided the control points to be able to regenerate the potential experimental geometry in a continuous fashion for numerical simu-

lations. This Bezier curve is consistent with the interpolated model coordinates provided by Ehrmann in his thesis (Ehrmann et al., 2017), where he has described the experimental model coordinates being interpolated directly from Abbott & von Doenhoff's geometry. The experimental pressure taps locations from the Sandia's LEES-Dataset (David et al., 2020) is also consistent with the Bezier curve.

Despite having very small differences between the geometries, results comparing the numerical simulations of both geome-

tries (Figure 7) has revealed that better validation is achieved using Abbot & von Doenhoff's original geometry, while the computed lift coefficients are slightly overpredicted with the Bezier geometry. Comparing the surface pressure distribution has shown that XFOIL and CFD results slightly under-predict the measured pressure values, irrespective of the geometry. It is noted that despite showing negligible differences between the computed pressure distributions for both geometries, for three different Reynolds numbers (Figure 7-10) the validation results with Abbot & von Doenhoff's original geometry shows good

agreement between the calculated and measured lift curve slope.

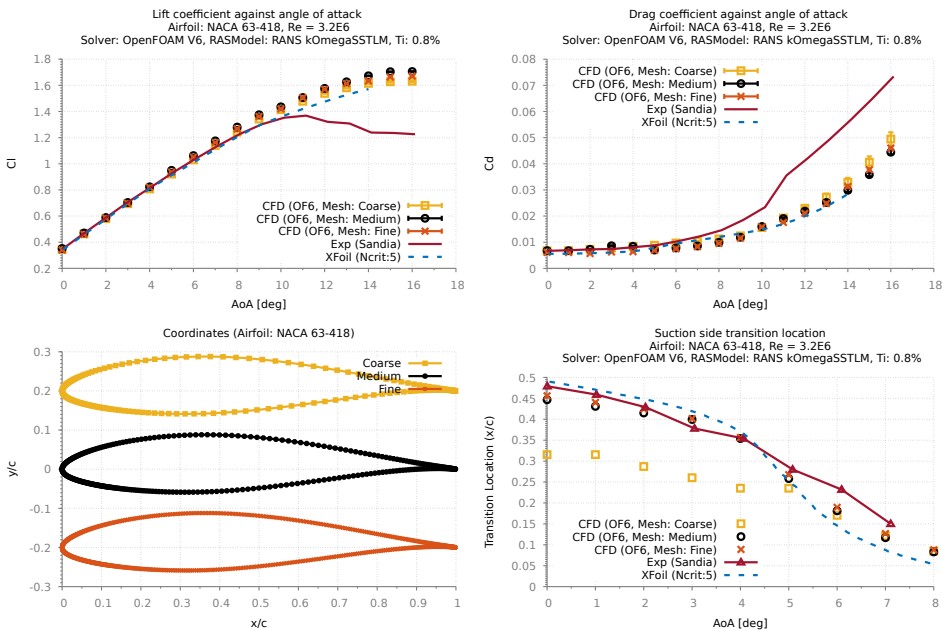

**Figure 5.** Comparing results for different grid densities

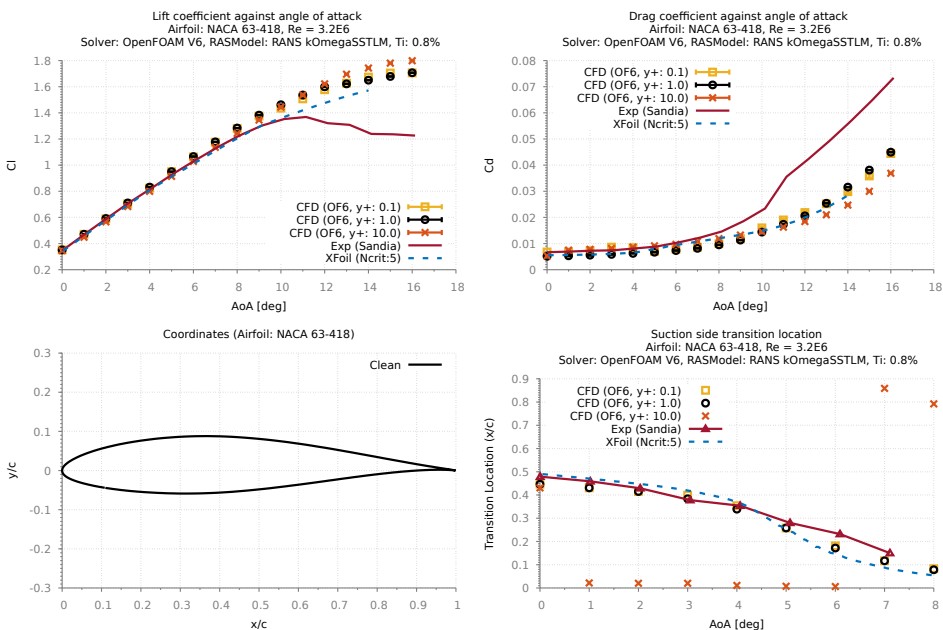

**Figure 6.** Comparing results for different y+ values





Nevertheless, the findings from both of the aforementioned test cases, make clear that the intended underlying SSTLM model is able to capture transition locations that agree well with the measurements, particularly for the Sandia experiment (David et al., 2020) using the Abbot & von Doenhoff's original or the Langel et al Bazier form of the NACA 63-418 geometry. However, to be consistent with the original experimental work as reported in (Ehrmann et al., 2017) (best measurement of the as-tested model) the Bezier geometry was chosen for calibrating the roughness amplification model.

It is also noted that this study was conducted using the SSTLM model originally implemented by Menter without the modification proposed by Langel et al. In order to reduce sensitivity to free stream turbulence values Langel et al (Langel et al., 2017b) have adopted the recommendation from Khayatzadeh et al (Khayatzadeh and Nadarajah, 2014) to use a different transition onset equation. This was accomplished by increasing the constant from 2.193 to 3.29 in the onset equation. However, the results from this verification study clearly show good agreement with the measured transition locations without this modification, thus the value of 2.193 originally implemented by Menter was used throughout this work.

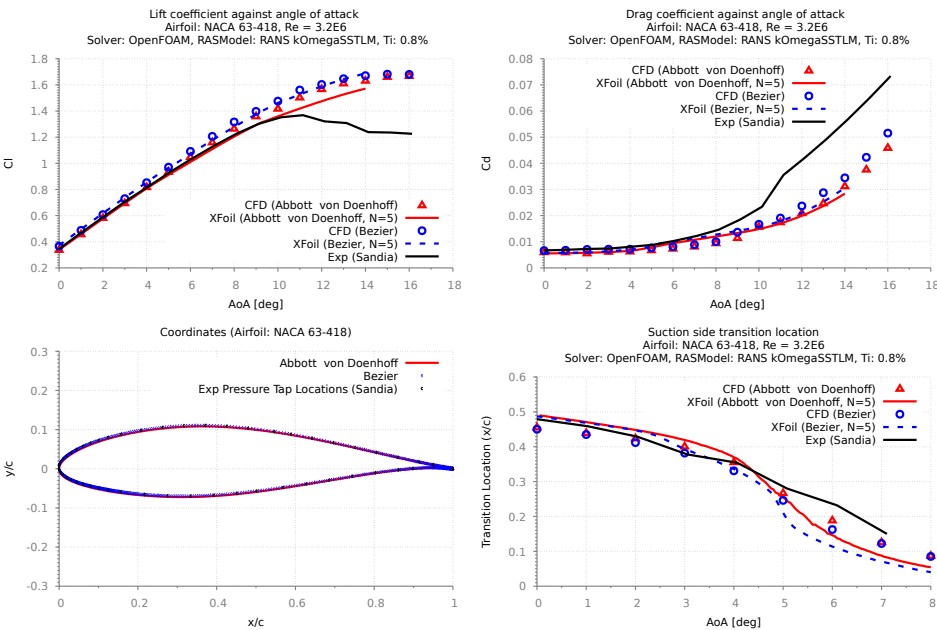

**Figure 7.** Comparing results for different NACA 63-418 geometries used to validate Sandia experiments

## 2.3 Flow transition prediction with rough surfaces - Verification and calibration

The detailed derivation of the model description for flow transition on rough surfaces is well documented by Langel et al (Langel et al., 2017b). Essentially, a new equation for $Re_\theta$ accounting roughness is defined based on the local roughness amplification quantity ($Ar$).

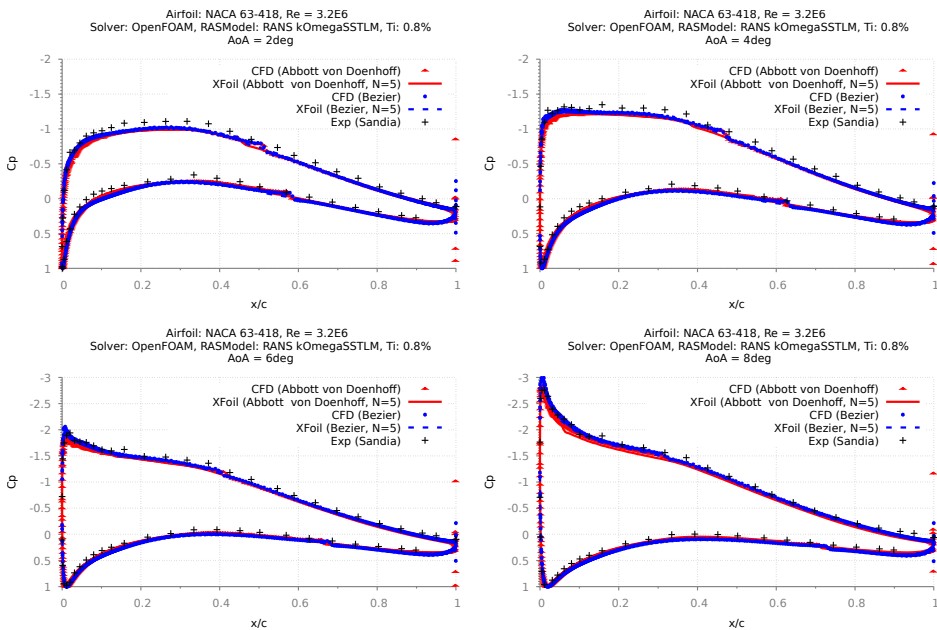

**Figure 8.** Comparing surface pressure distribution for different NACA 63-418 geometries used to validate Sandia experiments

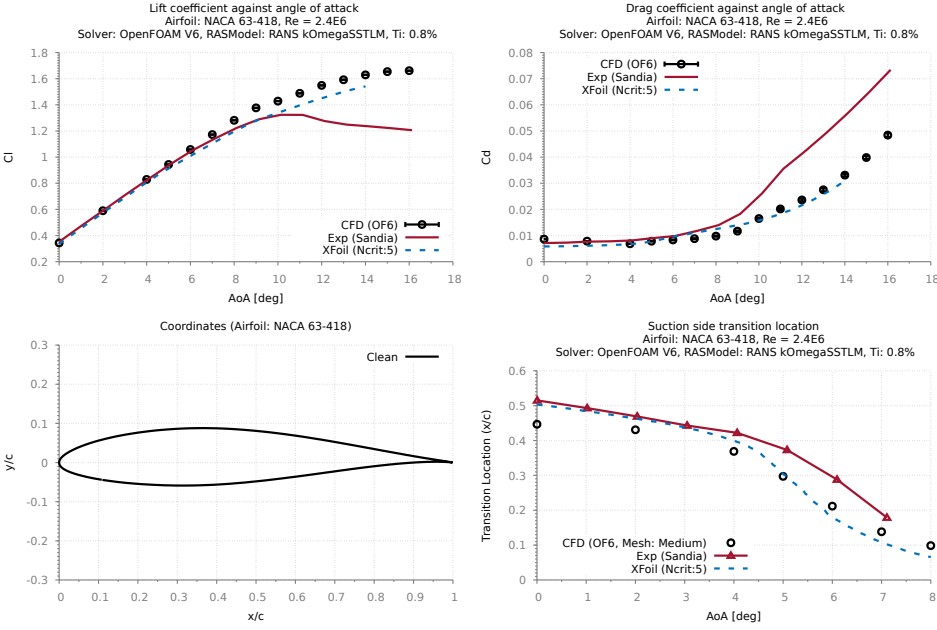

**Figure 9.** Validation results at Re= 2.4E6 using the Abbott & von Doenhoff's NACA 63-418



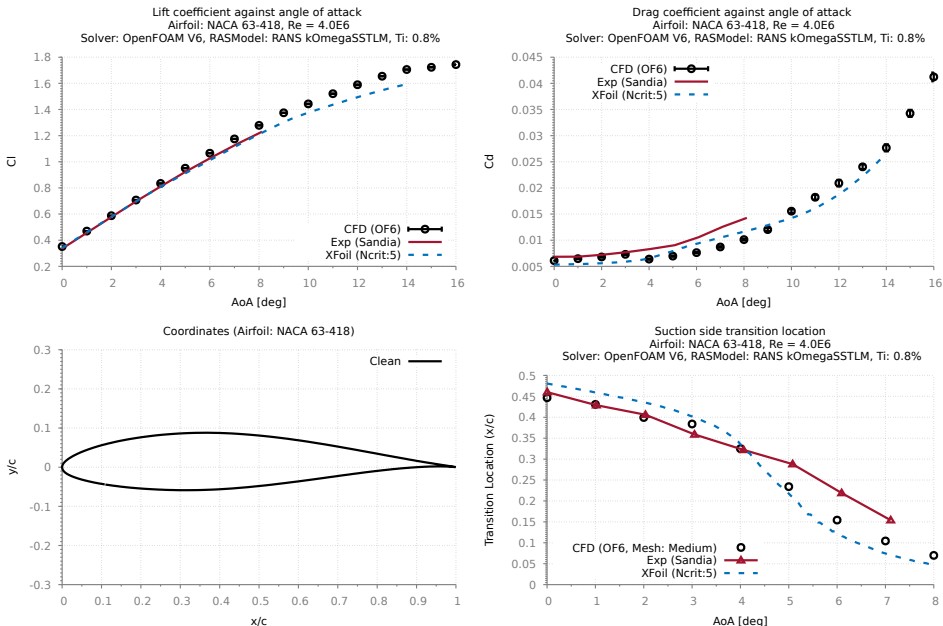

**Figure 10.** Validation results at Re= 4.0E6 using the Abbott & von Doenhoff's NACA 63-418

$$Re_{\theta,rough} = Re_\theta + \frac{1}{3}u^+ \left(\frac{Ar}{c_{r1}}\right)^3 - \frac{1}{2}\left(\frac{Ar}{c_{r1}}\right)^2 \tag{1}$$

Where $u^+$ is the logarithmic function for mean velocity profile, $c_{r1}$ is the model constant 8.0.

With the additional $Ar$ variable, one can choose to increase the local momentum thickness Reynolds number using the Eqn 1 or reduce the correlated critical value by a similar amount. Similar to Langel et al's implementation latter is considered by lowering the local correlation variable $\tilde{Re}_{\theta t}$ within the production term of its transport equation (Eqn 2).

$$\tilde{P}_{\theta t} = c_{\theta t}\frac{\rho}{t}\left[(Re_{\theta t} - \tilde{Re}_{\theta t})(1 - F_{\theta t}) - bF_{Ar}\right] \tag{2}$$

This implementation using the $F_{Ar}$ function serves to reduce the $\tilde{Re}_{\theta t}$ values downstream of the rough boundary, where elevated $Ar$ values are expected. With lower $\tilde{Re}_{\theta t}$, the $Re_\theta$ values required to trigger turbulence intermittency production are achieved easier, thus triggering transition with smaller disturbances. This approach allows onset transition to take place within the rough boundary itself, depending on the roughness height.

The transport equation for $Ar$ is defined similar to those of the underlying transition model, with a model constant $\sigma_{ar} = 10.0$:

$$\frac{\partial(\rho Ar)}{\partial t} + \frac{\partial(\rho U_j Ar)}{\partial x_j} = \frac{\partial}{\partial x_j}\left[\sigma_{ar}(\mu + \mu_t)\frac{\partial Ar}{\partial x_j}\right] \tag{3}$$





The $Ar$ parameter at the wall is a function of equivalent sand grain roughness height ($k_s$). In the past, several studies
have managed to correlate measured roughness parameters to an equivalent sand grain roughness heights, and Bons (2010)'
review article collates many of these correlations in one. Based on the correlated equivalent sand grain roughness height ($k_s$),
a distribution of $Ar$ is prescribed as a boundary condition following:

$$Ar|_{wall} = c_{r1}k^+ \text{ with } k^+ = \sqrt{\frac{\tau_w}{\rho}}\frac{k_s}{\nu} \tag{4}$$

The secondary cubic function (Eqn 5) with model constants $c_{r2}$ and $c_{r3}$, 0.0005 and 2.0 respectively are used at the lower
$Ar$ values to maintain small values for hydraulically smooth walls, where it has no effect on the transition model. A linear
function is switched at $C_{Ar} = \sqrt{c_{r3}/3c_{r2}}$ to matches the slope of the cubic function in order to prevent unphysical overshoots
at high levels of $Ar$.

$$F_{Ar} = \begin{cases} c_{r2} \cdot (Ar)^3 & : Ar < C_{Ar} \\ c_{r3} \cdot (Ar - C_{Ar}) + c_{r2}C_{Ar}^3 & : Ar \geq C_{Ar} \end{cases} \tag{5}$$

A blending function is also introduced to limit the process of reducing the production term of $\tilde{Re}_{\theta t}$ when its value nears the
prescribed minimum:

$$b = \left[ \frac{1}{2}sin\left( \frac{\pi}{155}\tilde{Re}_{\theta t} - \frac{97\pi}{155} \right) + \frac{1}{2} \right]^2 \tag{6}$$

Although the $Ar$ quantity allows for early transition triggering due to roughness, the effects on a fully turbulent boundary
layer and the lowering of turbulent dissipation rate ($\omega$) is also required to be adjusted in order to establish an accurate calculation
of the wall shear stresses. The following update is used for $\omega$ boundary condition at the wall:

$$\omega|_{rough\ wall} = \frac{\mu_\tau^2 S_r}{\nu} \text{ with } \mu_\tau = \sqrt{\frac{\tau_w}{\rho_w}} \text{ at wall} \tag{7}$$

where $S_r$ is based on the non dimensional sand grain roughness height ($k^+$):

$$S_r = \begin{cases} (50/k^+)^2 & \text{if } k^+ \leq 25 \\ 100/k^+ & \text{if } k^+ > 25 \end{cases} \tag{8}$$

### 2.3.1 Verification study (T3A)

Similar to the study conducted to verify the underlying SSTLM model, the T3A test setup was used to verify the working of the
implemented roughness amplification model (SSTLMkvAr) with its effect on modeling distributed roughness on the entire flat
plate. The results from this study (Figure 11) clearly show the forward movement of the transition location with increasing $k_s$.
It is also evident from the results, that the effect of roughness not only triggers early transition but as intended, is transported
downstream to account for larger wall shear stresses.



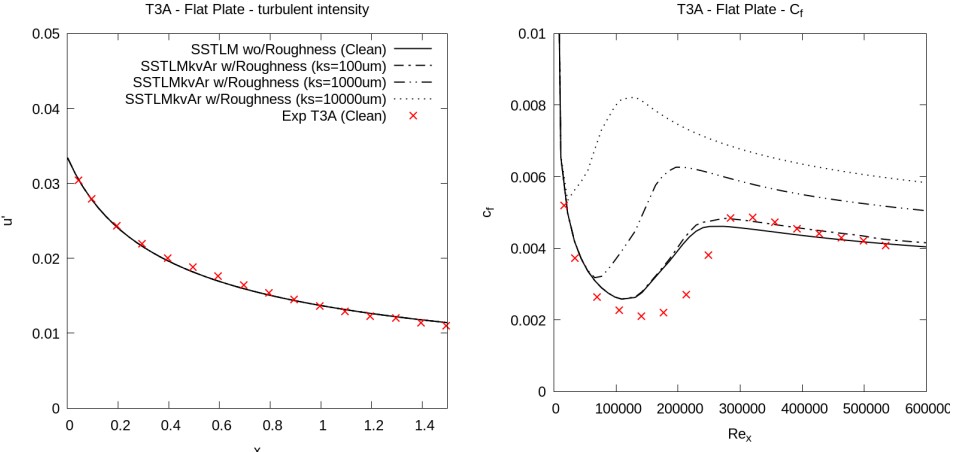

**Figure 11.** Turbulence kinetic energy and Wall shear stress along clean and rough flat plate

### 2.3.2 Model calibration

Ehrmann's experimental work (Ehrmann et al., 2017) documents the findings from testing various distributed roughness on the NACA 63-418 airfoil at three different Reynolds numbers (Re= 2.4E6, 3.2E6, and 4.0E6), the data from this work is published in the LEES dataset (David et al., 2020). Using the measured transition locations from this dataset, the newly implemented SSTLMkvAr model was calibrated for its sand grain roughness height ($k_s$). This calibration was performed for three different roughness heights: 100, 140, and 200um, at a constant roughness density of 3%. Furthermore, the calibration of three different

densities (3, 9, and 15%) was performed using the data for the roughness height of 100um. For both parameters, the calibration was performed at the flow Reynolds number of 3.2E6.

The calibration study was conducted by using a cost function that minimizes the errors between the calculated and measured transition locations across the whole experimented range of angles of attacks (-4 to 6°). As a result the following equation was established for $k_s$ as a function of roughness height ($R_h$) and density ($R_D$).

$$k_s = \left[ 1.49783 \times 10^9 \sqrt{9.29170684919976 \times 10^{24} R_h - 9.24939916631901 \times 10^{26}} - 2950.72 \right]$$
$$\times \left[ -0.768462 + 5.46075 \times 10^{17} \times \sqrt{3.16961301221313 \times 10^{32} \times R_D + 9.79025531790832 \times 10^{31}} \right] \quad (9)$$

Despite only calibrating the $k_s$ values to match the measured transition locations, the results show a very good agreement between the modeled and the experimented drag forces, especially for the large roughness heights of 140um and 200um (Figure 12 and 13). All modeled results showed notable differences (10%) with the measured lift forces, where the relative reduction in lift due to the rough leading edge was captured well, while the absolute values were over-predicted.

For the roughness height of 100um, the calibrated results showed reasonable agreement with the measured transition location. However, the corresponding results on the drag forces were under-predicted in comparison with the measured values.





Due to the lack of experimental data, it was only possible to study the effect of roughness density at the height of 100um. The results from this study (Figure 14 - 16) clearly show that the calibrated CFD model is able to calculate the forward movement of the transition point with increasing roughness density, while the corresponding effect on the drag forces is poorly predicted.

These results indicate that the calibrated CFD model fails to predict the measured aerodynamic forces for small LE roughnesses (100um).

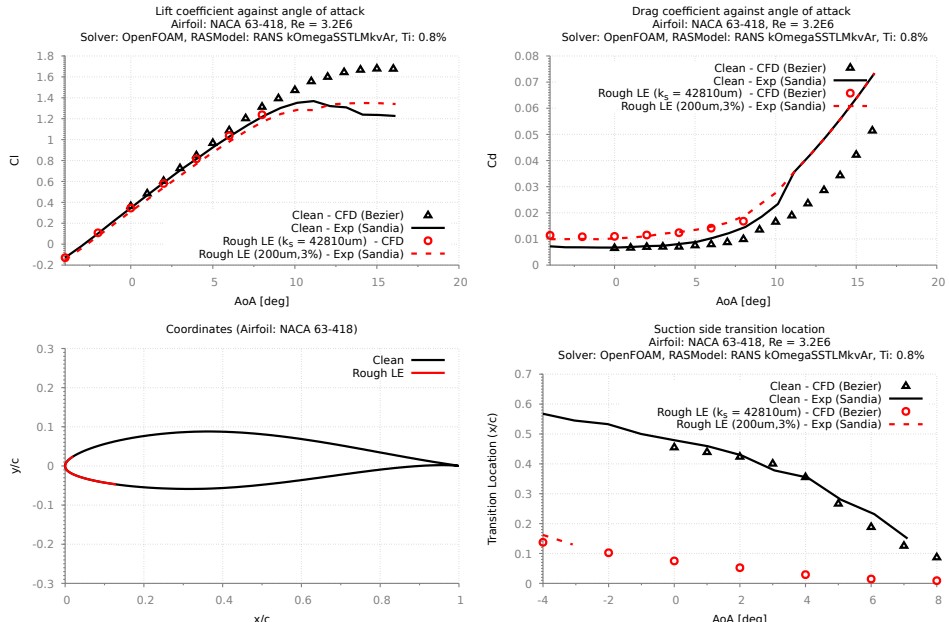

**Figure 12.** Calibration results at Re= 3.2E6 for NACA 63-418 with LE roughness height of 200um and 3% density

## 2.4 Discussion (Part 1)

An investigation to develop and implement a transport equation based boundary layer transition model for rough surfaces was successfully completed. This implementation was carried out for the open-source CFD code OpenFOAM, which was

then calibrated using an experimental study. Numerous verification and validation studies were performed to establish realistic results for predicting roughness induced flow transition for wind turbine applications.

Based on this study, it is clear that the newly developed CFD model, calibrated with the measured transition locations shows good agreement of the aerodynamic forces for airfoils with leading-edge roughness heights in the order of 140-200um. Despite matching the location of the transition point, the results indicate that for a smaller roughness height of 100um, the

model fails to accurately predict the measured forces. It is also noted that the resultant CFD model based on the limited amount of measurement data is strongly tuned to the Sandia experiment, using the NACA 63-418 at the flow Reynolds number of 3.2million, and its sensitivity to other airfoils at different flow Reynolds numbers are currently unknown. Thus, further

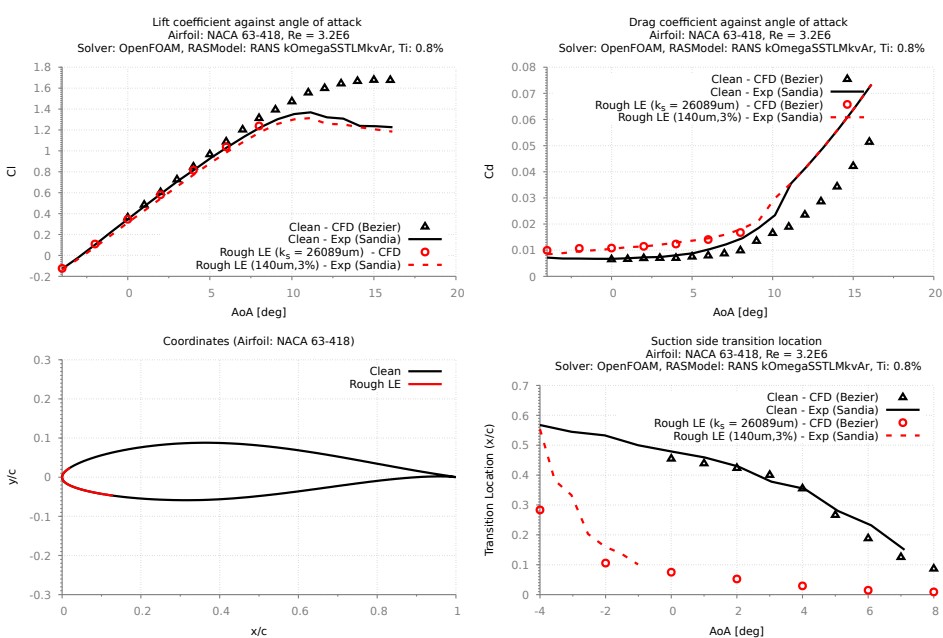

**Figure 13.** Calibration results at Re= 3.2E6 for NACA 63-418 with LE roughness height of 140um and 3% density

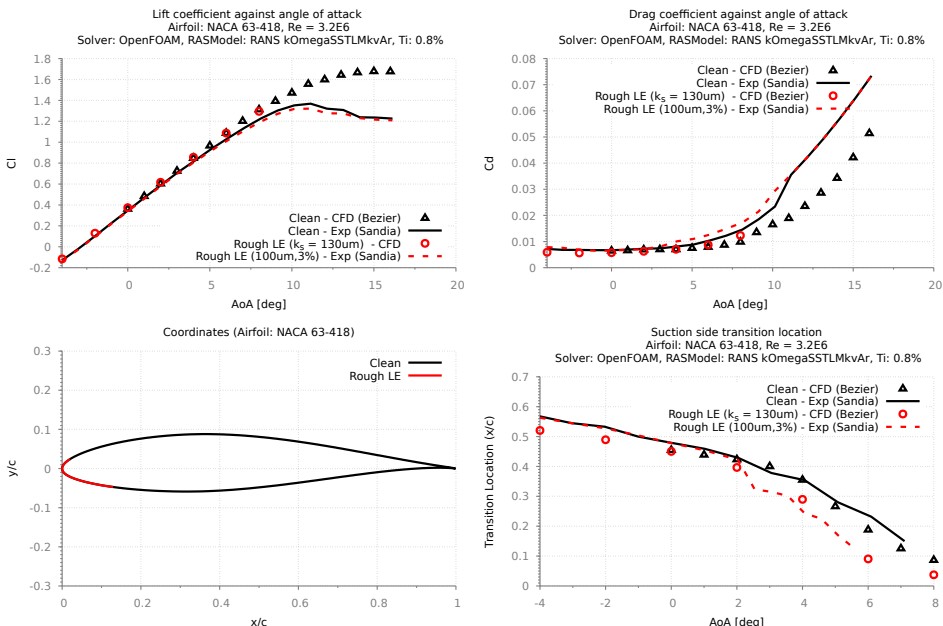

**Figure 14.** Calibration results at Re= 3.2E6 for NACA 63-418 with LE roughness height of 100um and 3% density



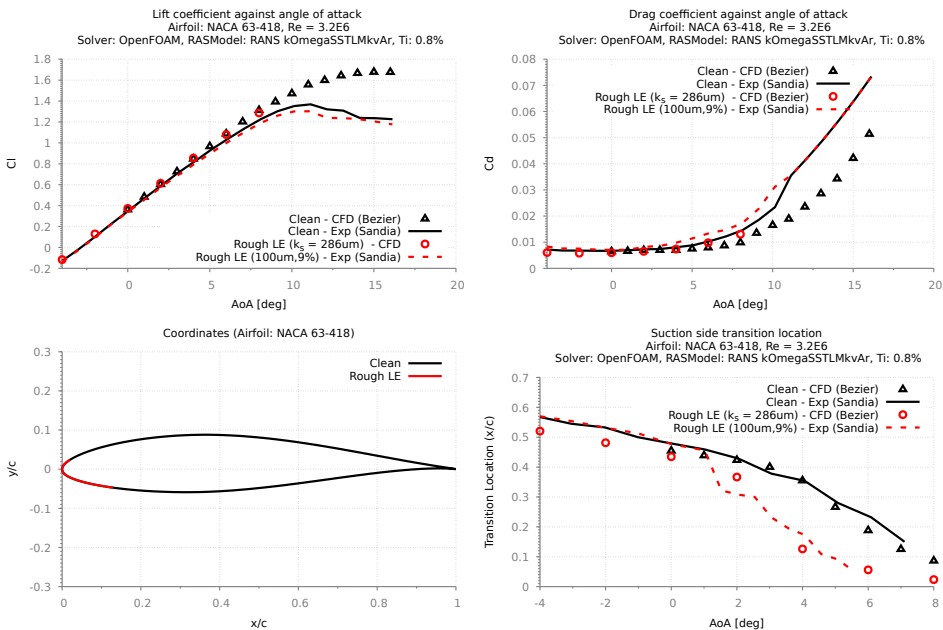

**Figure 15.** Calibration results at Re= 3.2E6 for NACA 63-418 with LE roughness height of 100um and 9% density

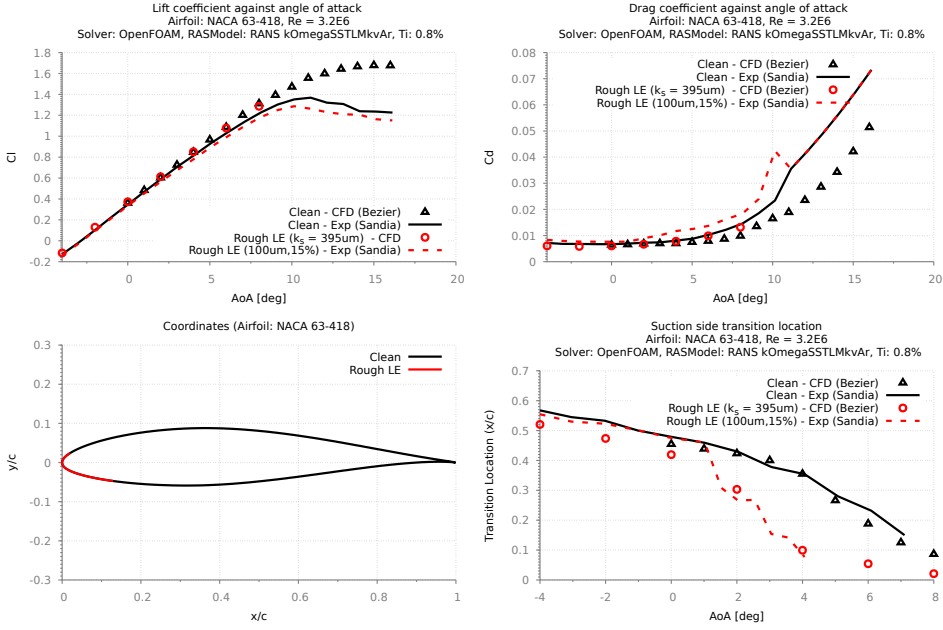

**Figure 16.** Calibration results at Re= 3.2E6 for NACA 63-418 with LE roughness height of 100um and 15% density



tuning with validation using independent measurement is highly recommended, especially for modeling the effect of roughness density.

## 3  Aerodynamic performance of 3D scanned eroded wind turbine blade (Part 2)

The largest wind turbine blades (WTBs) are deployed offshore, where noise restrictions are lower and therefore tip speeds can be higher. This leads to increased damage on the WTB by leading edge erosion (LEE, Figure 17), which is even a larger issue offshore than onshore in terms of expensive repair and maintenance (O&M) operations.

LEE has become a primary concern for the offshore wind industry. A detailed understanding of the mechanisms and prediction of LEE is one of the most important questions in developing wind turbine technologies and lowering the Levelized Cost of Energy (LCoE). Also, LEE of WTBs is one of the most common reasons for the reduction of power generation by wind turbines.

Relative small sized damages on the surface of the WTB have an impact on the overall performance and therefore the energy yield of a wind turbine. The LEE damage of wind turbines leads also to an increase in drag coefficient and decrease in lift coefficient for higher angles of attack. The tip part of the blade, due to the higher speeds, contributes to the majority of the torque production; the last 15% of the WTB radius for a multi-megawatt rotor can contribute up to 25% of its total power production. However, the higher speeds also result in higher droplet impact velocity and the erosion rate at this part of the blade.

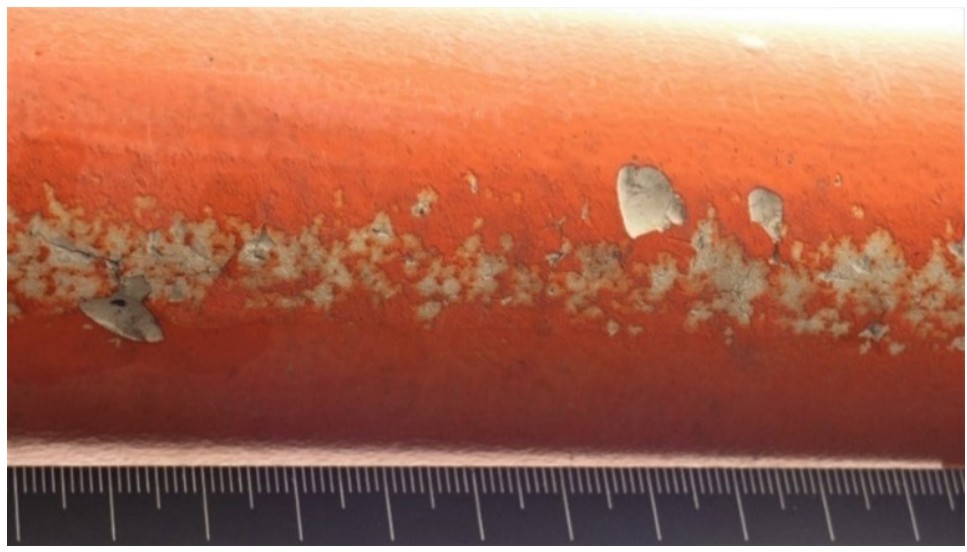

**Figure 17.** LEE damage on the tip section of a WTB, photographed by TNO

Detailed understanding of the impact on aerodynamic performance due to the eroded leading edge (LE) is required to optimize the repair and operation, and accurate modeling tools are therefore essential. Damage mechanisms of LEE, 3D





scanner technologies and Computational Fluid Dynamics (CFD) methods of modeling the erosion are discussed and presented. 3D images have added value for O&M purposes and quality control after repairing a WTB. Inspection and monitoring of a WTB are necessary to plan maintenance by keeping track of the degradation and structural health of the blade. Additional to visual inspection (manual or drone) the 3D scanning of the surface can be a suitable technique.

The novel aspect of this study is the modeling of high resolution LE surface of the WTB with LEE damage, that was captured using state-of-art optical high-resolution 3D scanner technologies. This makes this research different from numerically assumed blades with LEE damage.

Using the point cloud data, 2D and 3D Reynolds Average Navier Stokes (RANS) CFD models have been applied to investigate the influence of shape change at the LE on aerodynamic performance. Both models are shown to predict the change
in flow characteristics due to the LE differences. The results from CFD simulation have been used for BEM calculation to establish the performance of the WTBs in terms of power coefficient and relative changes in AEP. The AEP assessment was conducted using a mean wind speed of 10m/s with a Rayleigh wind distribution. A comparison of the simulation results is also made between two different CFD solvers.

The work was funded by TNO's Knowledge Innovation Program (KIP) for 2020.

**3.1   LEE mechanism**

LEE affects almost all wind turbines, and it has a harmful effect on the aerodynamic efficiency of the wind turbine blade, reducing the AEP, O&M cost, and the lifetime of the wind turbine. The energy losses associated with LEE on an operational wind farm are examined, with the average annual energy production dropping by 1.8% due to medium levels of erosion, with the worst affected turbine experiencing losses of 4.9% (Law and Koutsos, 2020).

Another study shows results of an in-depth study indicated that a heavily eroded WTB can reduce annual energy production by up to 5% for a utility scale wind turbine. Accounting for performance losses due to common blade surface roughness, such as dirt and insects, does not account for the increased performance loss due to the more severe surface roughness caused by the erosion of leading edge blade material (Maniaci, 2022).

Types of LEE and surface roughness (Slot et al., 2015) represented in Figure 18:

– 1) an incubation period, in which the surface is virtually unaffected. The roughness of the surface and small pit shaped defects in the coating will appear

– 2) the steady-state erosive wear stage where the surface wears at a relatively high wear-rate

– 3) the final erosion stage with a strongly reduced wear rate due to the higher surface roughness which was produced in the second phase. Damage reaches the structural composite material of WTB

That erosion has serious impacts on the rotor performance has been shown previously in a study at the University of Illinois where the DU 96-W-180 airfoil was tested in combination with a polyurethane wind protection tape. During the experiments, an increase in drag coefficient by 80- 200% was observed. Additionally, a decrease in lift coefficient for higher angles of attack



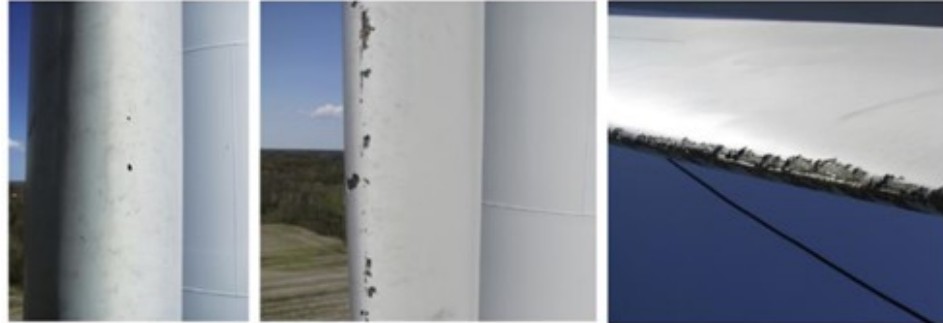

**Figure 18.** Changes to the LE surface due to impact with a different degree of severity, From left: LEE at an early stage, severely damaged coating and LEE starts to wear the shell material. Adapted from Keegan et al. (2013); Rempel (2012)

was ascertained. The results also implied a significant loss in annual energy yield by 7% for an increased drag coefficient of 80%, which underlines the necessity of an independent, comparable and repeatable erosion test (Liersch and Michael, 2014).

The NREL 5 MW turbine is simulated with clean and eroded blades, which are compared to coated blades equipped with leading edge protection. Aerodynamic polars are generated by means of Computational Fluid Dynamics, and load calculations are conducted using the blade element momentum theory. The analysis in this work shows that, compared to clean rotor blades, the worse aerodynamic behavior of strongly eroded blades can lead to power losses of 9%. In contrast, coated blades only have a small impact on the turbine power of less than 1% (Schramm et al., 2017).

**3.2   3D scanning technology for WTBs**

Periodic replacement and maintenance of damaged WTBs result in significant downtime and operational costs. Mainly the costs associated with the periodic repair of the blade coating and tape or even replacement of the complete WTB. The selection of the most suitable type of LEP (coating, tape, shell) depends on the conditions to which the WTB is exposed during the operational period (precipitation type and intensity, tip speed, UV, temperature). The current situation is that a maintenance-free WTB

design is not available and the high cost of the blades made it necessary to find new methods for inspection, monitoring, and repair.

There are no known references for optical scanning systems that are already used in practice for O&M of WTBs. Several innovative solutions have been developed for scanning WTBs in a controlled environment for production quality control. Amongst them, Aanæs et al. (2018) has described a complete autonomous inspection system for WTB surfaces. The mea-

surement system presented here is composed of a high-resolution, structured light-based 3D scanner and a local positioning system. The locomotion system is composed of a six-axis industrial robot mounted and a moveable platform, and this design is described in the following. It was found that a combination of a structured light 3D scanner and a laser tracker with an active target formed a good measurement system and that the range of this system could be extended by an industrial robot mounted



on a moveable platform. FORCE Technology and SGRE developed an optical scan application with an autonomous robot for
quality control of the WTB after production Jensen (2019).

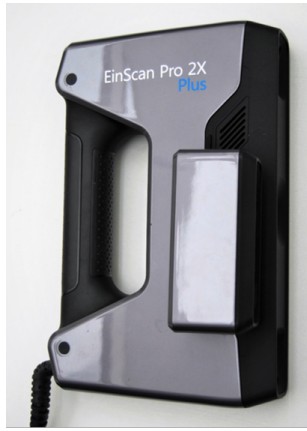
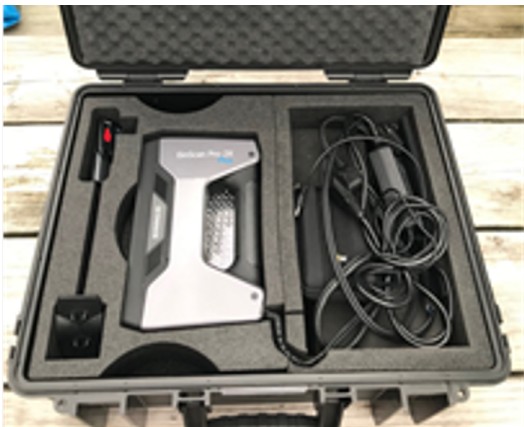

**Figure 19.** EinScan Pro 2X Plus Handheld 3D Scanner and the flight case with the equipment, photographed by TNO

Different 3D hand scanning tools are on the market available and development is going rapidly for different practical applications like scanning large objects like WTBs for production quality purposes Shining3D (2018); Aanæs et al. (2018) and also applications in outdoor conditions. For this application the Shining3D EinScan Pro 2X Plus handheld 3D scanner is selected based on the performance, ease to use, and good price-quality relation (Figure 19).

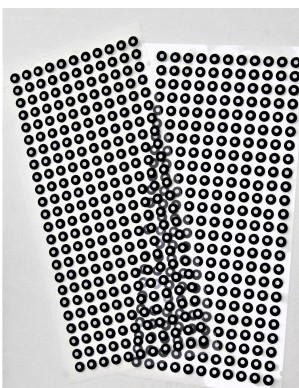
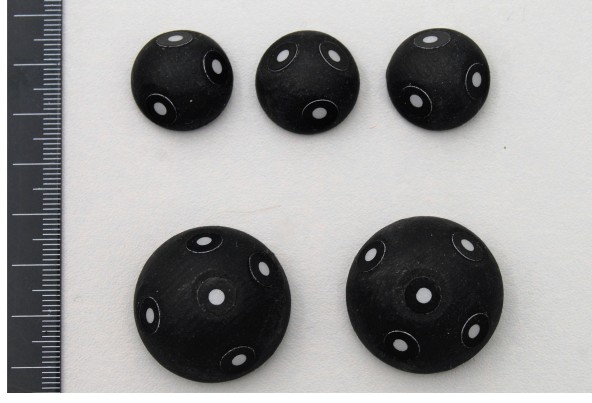
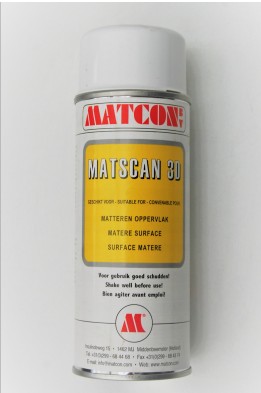

**Figure 20.** From left to right: 3D marker stickers, large spherical markers, 3D scan anti reflection spray, photographed by TNO

For the scanning of a WTB profile the handheld rapid scan mode is applicable. For detailed scanning of a damage profile, like LEE damage, the handheld HD scan mode is recommended with maximum scanning accuracy of 0.05 mm. The HD scan mode requires the application of 3D markers on the surface (Figure 20). The Prima pack add on camera (Figure 21) improves the capability to scan objects like a WTB surface without surface morphology and markings like 3D marker stickers, holes, ridges, lines, edges, etc. The experiments are performed by TNO in the field with different used WTBs and different conditions.



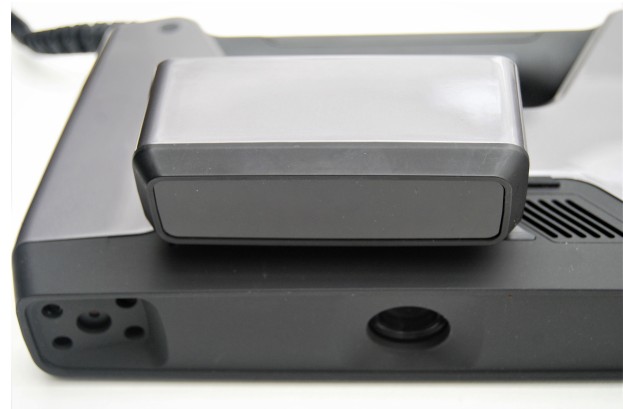

**Figure 21.** Prima pack add on plugged in the handheld scanner, photographed by TNO

The EinScan Pro 2X Plus Handheld 3D Scanner is an optical scanner that restricts the usage environment. Too strong ambient light outdoors will affect the scanner light, so that the light projected from the scanner will be hard to capture the data properly. Therefore, it is recommended to choose cloudy days or dusk when the light intensity has been reduced when scanning outdoors. For practical situations and problems, possible measures and solutions are given in Table 1.

### 3.3   3D scanning of a WTB and conversion to CAD format

The applied 3D scan is performed at location Eemshaven, the Netherlands on a research turbine with a blade radius of 63m. The scan operation is performed in cloudy conditions with low ambient light intensity. For the high-resolution scan mode (0.05 mm resolution) the 3D marker stickers were applied on the surface (Figure 22).

For scanning objects with large curved surfaces, like a WTB, the fast handheld scan mode with the medium of low resolution is recommended in combination with the Prima Pack add on camera (Figure 21). When the scanner does not recognize the 305 surface characteristics (texture, color variation) for tracking marker stickers can be applied. For large areas (> 1 m2) multiple scans are necessary which must be merged in the software tool into a single point cloud image. Large 3D markers (Figure 20) support the scanning and merging of the large sections.

An example of the WTB tip scanned with the fast scan mode without application of markers is shown in Figure 23.

The generated 3D point cloud is imported into the Geomagic Essentials 1 software tool (Figure 24) and processed with the 310 "Mesh doctor" tool to remove holes (3D markers), imperfections, and noise. The "watertight" mesh model (Figure 25) must be transformed into a CAD format, *.IGS or *.STL, for application in CFD.

The *.STL file format is commonly used for digitalization, smoothing, and re-meshing of the 3D models. The imported database file (point cloud) of the scanned sections of the blade should be translated in *.SLDPRT file format. Then a new geometry file can be converted to *.IGS or *.STL format and imported to Pointwise meshing tool.





**Table 1.** Limitations and mitigating measures for scanning in outdoor conditions

| Situation | Problem | Mitigating action |
|---|---|---|
| Scanning in direct sunlight | Scanning of the object is not possible | -Shield the object from the sunlight with a photo parasol or shelter<br>-Select low light intensity conditions like cloudy, early morning (sunset), or evening (dawn) times for the scanning<br>-Check the light intensity (Lux) on the object with a Lux measurement device |
| Shadow areas on the surface | Scan software recognizes the shadow areas as holes in the surface | -Adjust brightness during the scanning operation<br>-Perform separate scans (save as different projects) and merge in the software tool |
| Shiny surface with reflection of light | No tracking is possible with no recognition of surface texture | -When surface is wet first dry the surface<br>Apply 3D powder spray on the surface (Figure 20) |
| Large surfaces with no texture | Difficult to track the position during scanning | -Apply 3D marker stickers (Figure 4). Minimal number of 5 in a A4 area,<br>-Apply larger markers on the surface to make it easier to locate and merge scan sections |

### 3.4 CFD modelling of scanned LE

Several 2D and 3D numerical simulations were conducted to model the scanned surface of the eroded blade section. Two different steady Reynolds Average Navier Stokes (RANS) turbulence models were investigated for their relative differences in modeling eroded blades. One considers a fully turbulent boundary layer using Mentors SST turbulence model (Menter, 1994), while the other includes modeling of the laminar-turbulent flow transition using $\gamma - Re\theta_t$ correlation based SST model (SSTLM) Langtry (2006). Both turbulence models were validated on the DU97-W-300 airfoil (Figure 26) at the flow Reynolds number of 2E6, based on the experimental data from Baldacchino et al. (2018).

#### 3.4.1 Validation study

The results from the validation study showed that both SST (Figure 27) and SSTLM (Figure 28) RANS CFD model captures the aerodynamic performances of a wind turbine section with reasonable accuracy at pre-stall (attached flow) conditions.





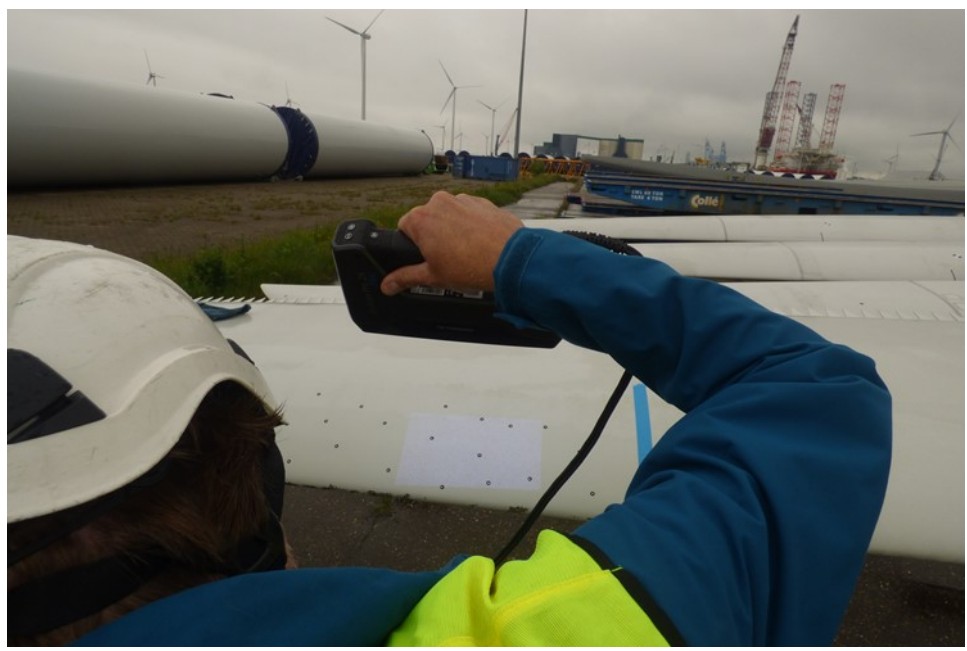

**Figure 22.** 3D scanning of a tip section of a WTB using sticker markers, photographed by TNO

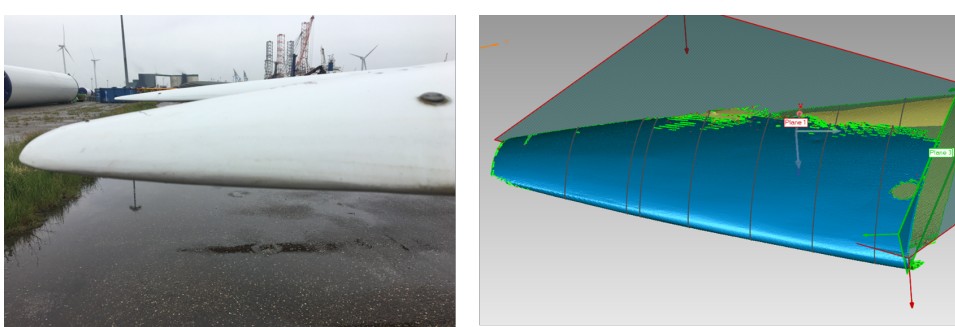

**Figure 23.** WTB tip (left, photographed by TNO) and the corresponding 3D point cloud image generated by TNO using the Geomagic Essentials tool.

A detailed validation study was conducted using surface pressure measurements for both turbulence models (Figure 29 and30), based on the result it was also found that both CFD models concurrently calculate surface pressures that are in good agreement with the attached flow measurements.

    The validation results indicate that the investigated CFD model is able to model the flow physics associated with fully turbulent boundary layer and flow transition and replicate the trends in lift and drag forces.

In comparison with experimental measurements, the computed clean airfoil lift and drag forces were validated within 10% of the measurements (AoA: 0 to 7.5deg). Unsteady computations may improve the validation at the large angles of attack.



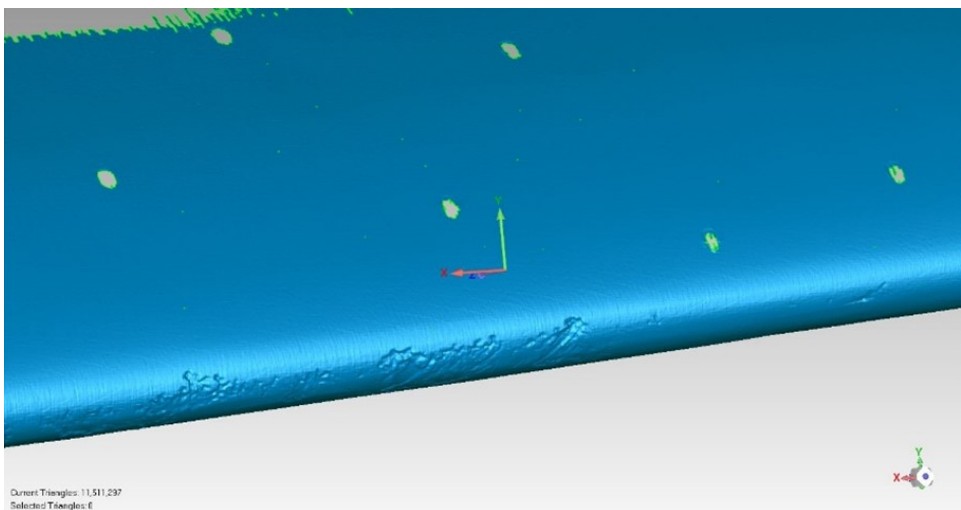

**Figure 24.** Screenshot generated by TNO of the meshed 3D point cloud imported in Geomagic Essentials 1 software tool. Green holes on the surface are the locations of the 3D sticker markers

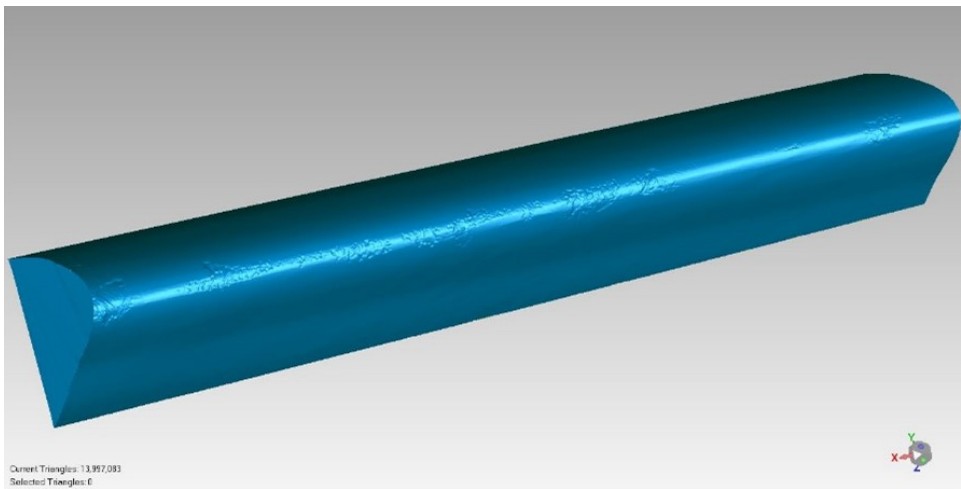

**Figure 25.** Screenshot generated by TNO of the processed watertight mesh model of the scanned LE

Nevertheless, based on the results it is clear that the relative changes in surface pressure and the trend in lift and drag forces for differing boundary layer conditions are captured with the chosen CFD model.

### 3.4.2 Results from modelling 3D scanned surface

Using the validated CFD model a smaller portion of the scanned surface containing the worst part of the blade was discretized for the numerical simulation. A 3D study was performed to analyze the impact of erosion as scanned (Figure 31), in parallel





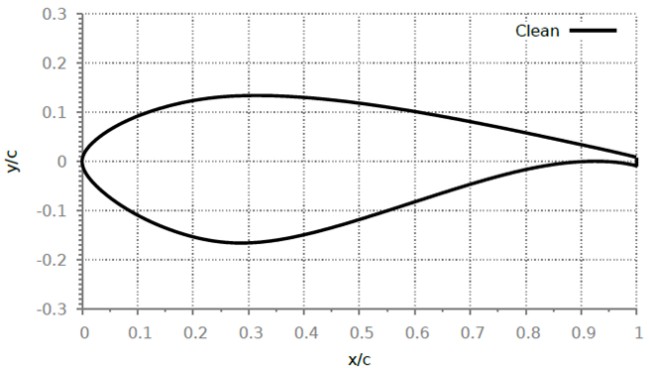

**Figure 26.** DU97-W-300 airfoil coordinates

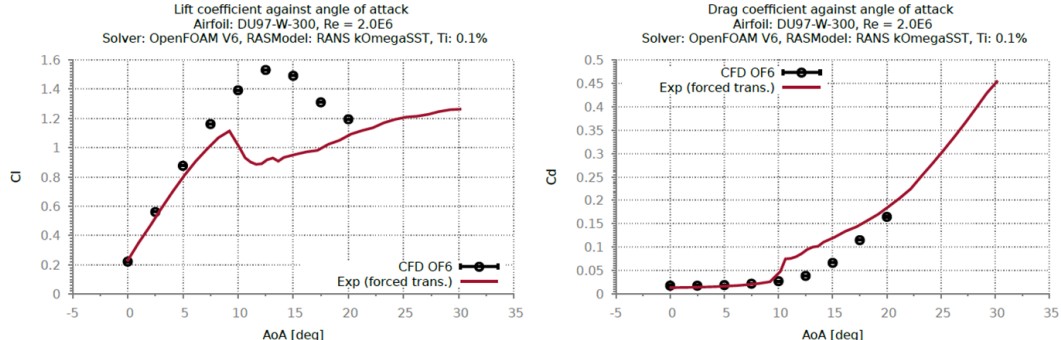

**Figure 27.** Lift and Drag coefficient against angle of attack validation against experimental fully turbulent boundary layer (SST)

2D sensitivity study was also conducted using the worst eroded cross-section within this part to establish a correlation map between LEE against AEP.

For the 3D simulations, two different grid densities were tested for grid independent results. The first grid was constructed with 3 million hexahedral elements, with a resolution of 0.2mm at the leading edge, while the fine grid was generated with 10 million hexahedral elements with a resolution of 0.1mm at the leading edge. The clean blade section was also modeled with 3 million elements for comparison.

The results from the 3D simulations clearly show detailed pressure resolution across the degraded surface (Figure 32), however, overall results show a negligible difference in the aerodynamic performance (Figure 33) is expected under a fully turbulent boundary layer. It was also noticed that despite having inflicted local flow disturbances at the leading edge, the eroded surfaces showed negligible effect on the flow downstream.





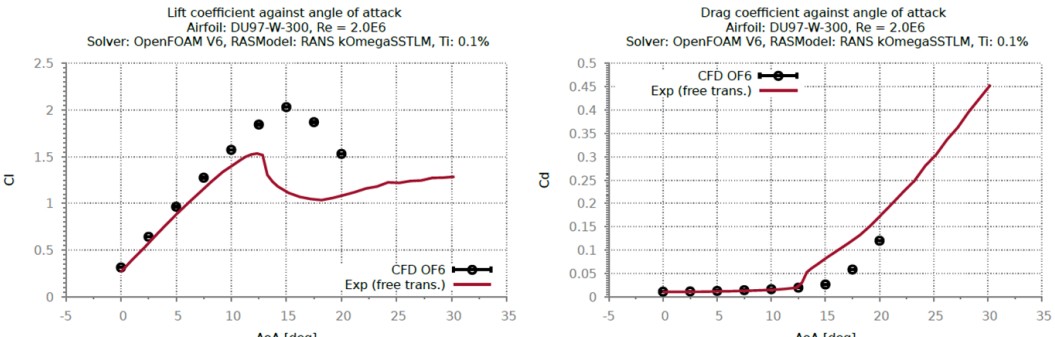

**Figure 28.** Lift and Drag coefficient against angle of attack validation against experiment – with boundary layer transition (SSTLM)

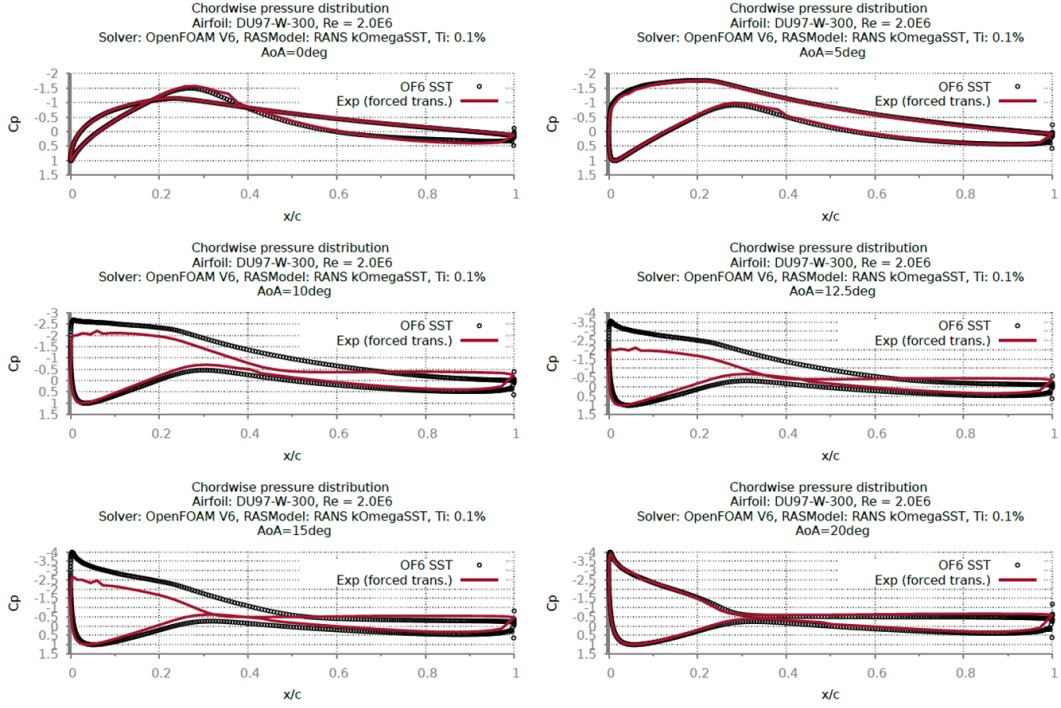

**Figure 29.** Chordwise pressure distribution comparison with experiment - forced transition/fully turbulent boundary layer (SST)

### 3.4.3 Results from 2D Sensitivity study

Several 2D CFD studies were conducted to study the impact of LE shape change due to erosion. This study was conducted by extracting a 2D cross-section from the scanned surface (Figure 34) and then generating two more profiles by translating the

extracted profile towards the trailing edge based on the distance of the material removed from the LE.

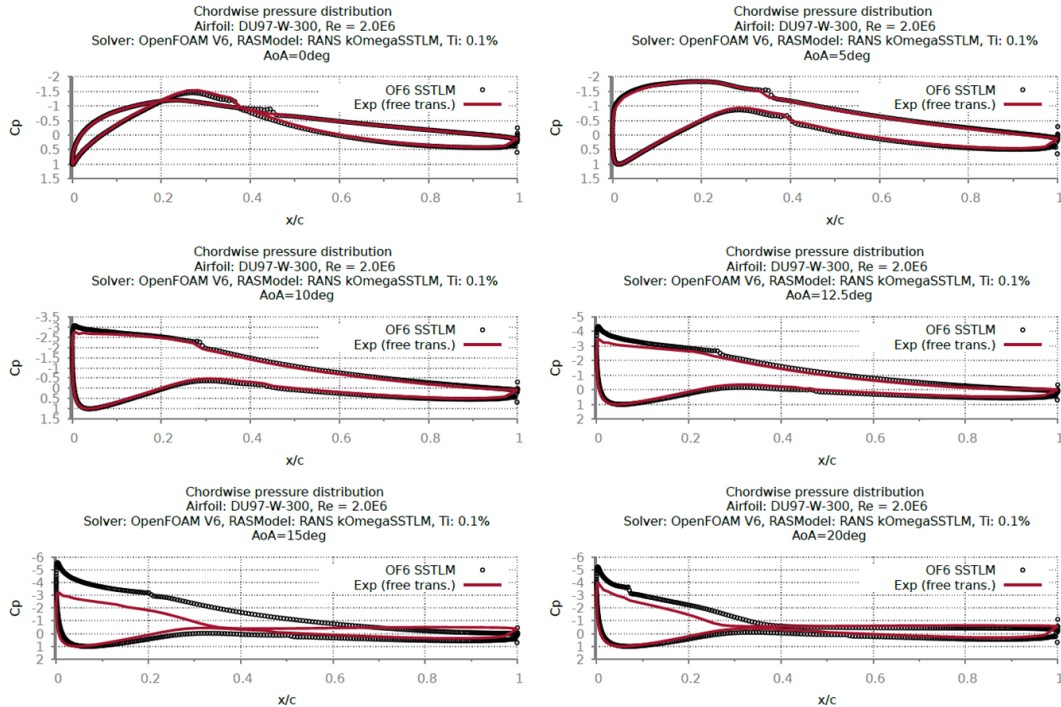

**Figure 30.** Chordwise pressure distribution comparison with experiment - free transition boundary layer (SSTLM)

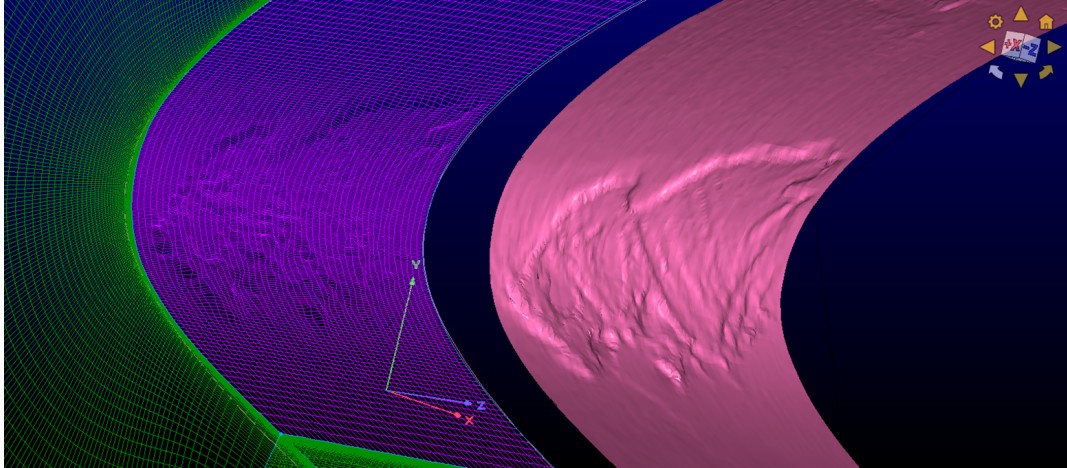

**Figure 31.** Selected portion of the scanned LE surface and its numerical grid

The relative shape changes measured and translated were superimposed (Figure 35) on a generic WTB tip airfoil NACA64-618, specifically the NREL 5MW reference turbine.

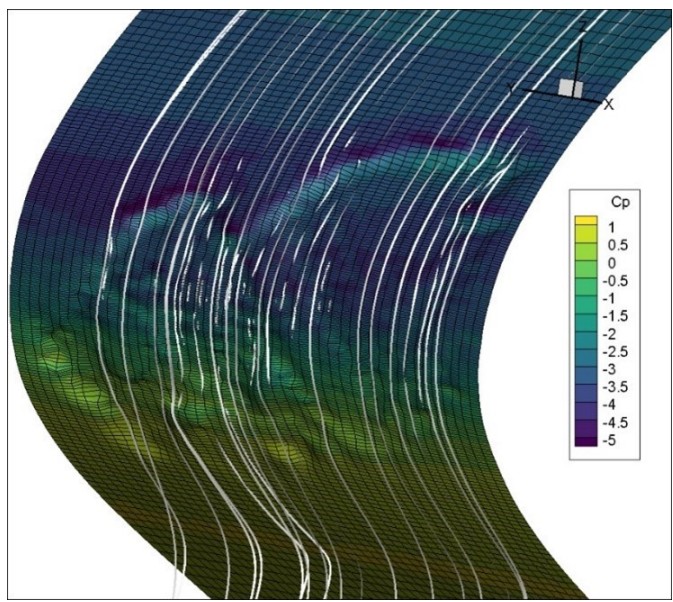

**Figure 32.** Pressure contours with flow ribbons at the eroded leading edge (AoA=10deg, SST, Grid Res. 3Mil)

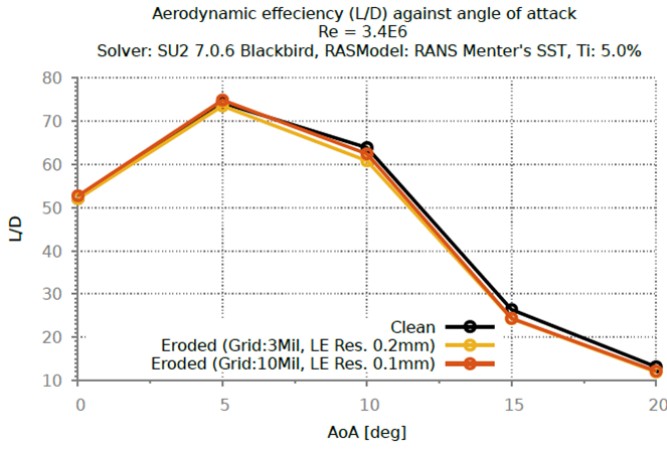

**Figure 33.** Lift to drag ratio comparison between clean and eroded blade section

The results from this investigation also suggest that under fully turbulent condition the investigated LE shapes show the least amount of influence on the aero performances (Figure 36) and results in a negligible difference to AEP of the NREL 5MW
rotor (-0.02%, Eroded 3x up to the last 30% of blade radius).

Under a transitional flow environment, results show a much larger aerodynamic impact due to the investigated shape changes at the LE. The results show up to 50% reduction in aerodynamic efficiency (Figure 37) even at the lower angles of attack, when the shape of the LE degrades by 0.8% of the chord (Eroded 3x). Based on transition modeled results, the BEM investigation





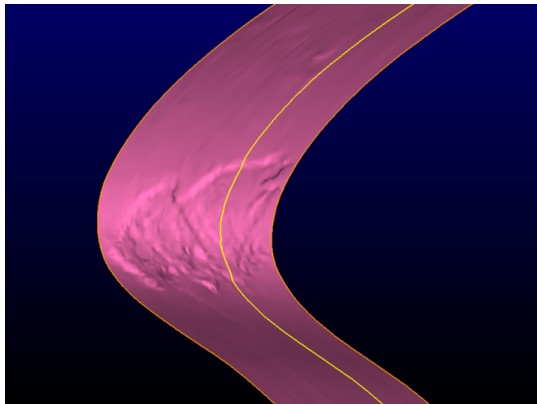

**Figure 34.** Extracted profile for 2D sensitivity study

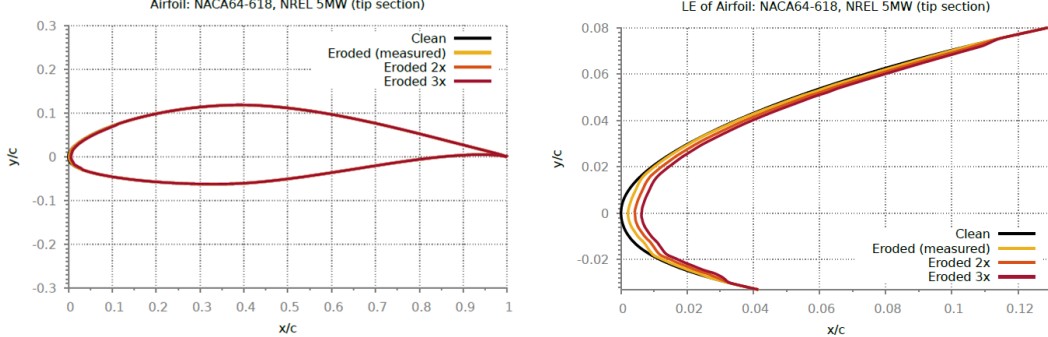

**Figure 35.** Superimposed eroded LE change for NACA64-618

examining different span extent (Table 2) of erosion has revealed up to 0.86% reduction in AEP when the LE shape is changed
by 0.8% of the chord for the last 30% of the blade span. However, if a lower mean wind speed is considered for the wind
distribution, the AEP loss is calculated to be much higher (Figure 38). This clearly indicates that the relative AEP loss from
erosion is much higher for turbines with a low capacity factor and vice versa for a high capacity factor as the turbines operate
at mostly above-rated conditions, where there is no loss from erosion.

## 3.5    Discussion (Part 2)

Based on the research conducted with the EinScan Pro 2X Plus Handheld 3D Scanner, the method has been shown to be
suitable for generating the 3D point cloud data input of WTB surfaces for CFD analysis. There are limitations to outdoor
scanning conditions where direct sunlight and shadows on the surface will disturb or make scanning of the surface with this
LED scanner impossible. Mitigating measures include shielding from direct sunlight (shelter or parasol) and scanning at low
light intensity, such as sunrise, sunset or cloudy conditions. The 3D scan method can be applicable for O&M on WTBs. In



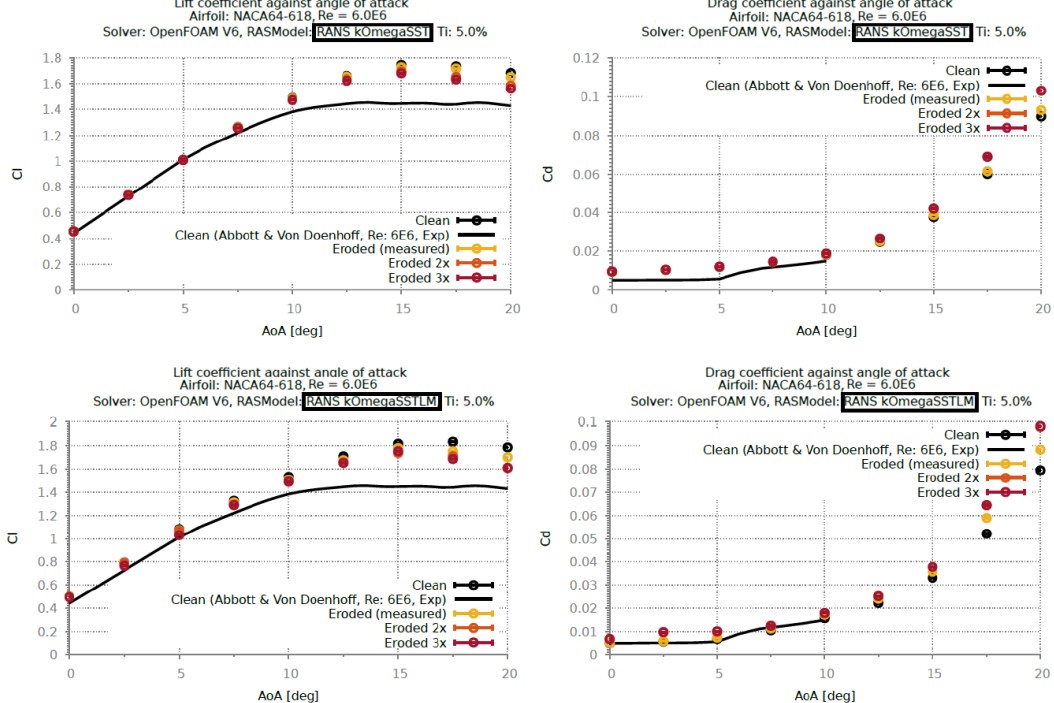

**Figure 36.** 2D Lift and drag coefficient sensitivity against angle of attack for NACA64-618 section (top: SST, bottom: SSTLM)

**Table 2.** Relative change (compared to SSTLM Clean) in AEP of NREL5MW rotor with Rayleigh 10m/s mean wind speed, under transitional condition considering different degree of LEE for different tip span extent

|  | Span extent from tip [%] | | |
| --- | --- | --- | --- |
|  | 30 | 20 | 10 |
| Eroded (measured) | -0.11% | -0.09% | -0.05% |
| Eroded 2x | -0.12% | -0.10% | -0.06% |
| Eroded 3x | -0.86% | -0.71% | -0.38% |

addition, to inspect and monitor LEE damage, other types of failure mechanisms can also be included that result in changes in the WTB profile. Examples of WTB damage are sand erosion, surface cracking and lightning strike damage.

The CFD models have clearly shown to predict the change in flow characteristics due to the LE differences. The results from CFD simulation have been used for BEM calculation to establish the performance of the WTBs in terms of power coefficient and relative changes in AEP. Comparisons of the simulation results were also made between two different turbulence models.

Based on the results it was found that under fully turbulent flow conditions the investigated shape difference at LE has negligible differences in aerodynamic performance, however, when flow transition is modelled, the results show a significant impact on



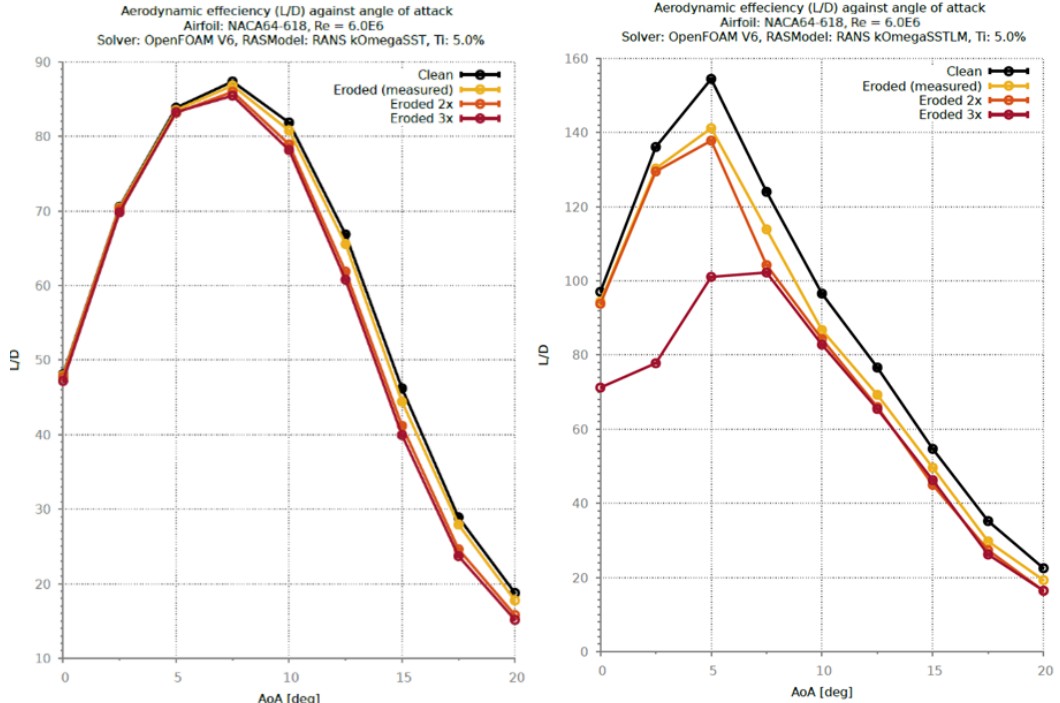

**Figure 37.** 2D Lift to drag ratio sensitivity against angle of attack for NACA64-618 section (left: SST, right:SSTLM)

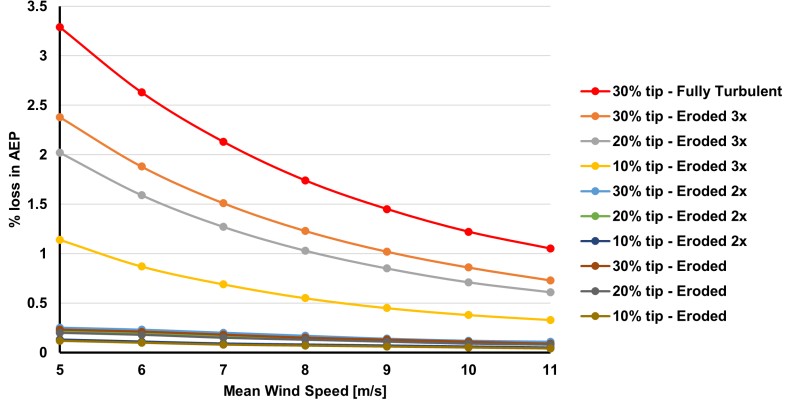

**Figure 38.** Percentage loss in AEP of NREL 5MW turbine considering different degree of LLE for different tip extent

aerodynamic efficiency. BEM investigation examining different span extent of erosion for the NREL 5MW rotor has revealed that up to 0.86% reduction in AEP can be expected when the LE shape is changed by 0.8% of the chord. It is noted that the aerodynamic modelling adapted for this work does not account for differences due to surface texture and has only modelled

the shape change due to LEE, and if considering the effect of flow transition being trigged by the surface textural difference





due to LEE then a larger reduction in AEP was calculated (-1.24%). It is also noted that if the turbine operates mostly in the sub-rated region of the power curve or is placed at a low wind site (low capacity operation), a much larger AEP loss (-3.3%) is realised due to LEE.

Although the generated grid resolution was made to capture numerical details in the order of 0.1mm, the RANS turbulence modelling approach used in this study is inherently flawed for capturing flow details at this scale of changes in geometry. The steady RANS modelling was chosen due to its computational efficiency and to meet the project time constraints. Despite RANS being considered an industry standard and quite accurate for many flow problems, the results may differ when a higher order turbulence model such as LES or Hybrid RANS-LES models are used to model these fine geometrical details.

Overall, this investigation to evaluate the aerodynamic impacts due to actual LEE has been successfully completed. Based 390 on the results, it was established that under the assumptions of this work, considering pure shape changes in LE of the tip part of the blade for a 5MW wind turbine can reduce its AEP in the order of 3.3%. However, if the effect of the exposed fibrous blade material on the transitional boundary layer was included using a model like the one described in Part 1, it will further deplete the sectional L/D ratios, thus resulting in further loss in AEP.

## 4 Conclusions

Leading edge erosion (LEE) is one of the most critical degradation mechanisms that occur with wind turbine blades (WTBs), generally starting from the tip section of the blade. A detailed understanding of the LEE process and the impact on aerodynamic performance due to the damaged leading edge (LE) is required to select the most appropriate Leading Edge Protection (LEP) system and optimize blade maintenance. Providing accurate modeling tools are therefore essential.

Based on this two-part study investigating different order of magnitudes in erosion damage has shown that up to 0.86-1.24% 400 reduction in AEP can be expected when the LE shape is degraded by 0.8% of the chord, based on the NREL 5MW turbine considering 10m/s mean wind speed Raleigh wind distribution. However, for low capacity operations such as operating mostly in the sub-rated region of the power curve or being placed at a lower mean wind speed site, a larger reduction in AEP is calculated due to LEE (2.4-3.3%). The results also suggest that under the fully turbulent condition the investigated LE shapes show the least amount of influence on the aerodynamic performances and result in negligible difference to AEP. Present CFD 405 model for modeling flow transition accounting roughness shows good agreement of the aerodynamic forces for airfoils with leading-edge roughness heights in the order of 140-200um while showing poor agreement for smaller roughness heights in the order of 100um. It is also noted that the resultant CFD model based on the limited amount of measurement data is strongly tuned to the Sandia experiment, using the NACA 63-418 at the flow Reynolds number of 3.2million, and its sensitivity to other airfoils at different flow Reynolds numbers are currently unknown. Thus, further tuning with validation using independent 410 measurement is highly recommended, especially for modeling the effect of roughness density.



**Appendix A: Acronyms**

| | |
|---|---|
| AEP | Annual energy production |
| BEM | Blade element momentum |
| Cd | Coefficient of drag |
| CFD | Computational fluid dynamics |
| Cl | Coefficient of lift |
| LCoE | Levelized cost of energy |
| LE | Leading edge |
| LEE | Leading edge erosion |
| LEP | Leading edge protection |
| LES | Large eddy simulation |
| O&M | Operations and maintenance |
| RANS | Reynolds averaged Navier Stokes |
| Re | Reynolds number |
| SST | k-omega shear stress transport fully turbulent turbulence model |
| SSTLM | k-omega shear stress transport with Langtry-Menter transition turbulence model |
| SSTLMkvAr | k-omega shear stress transport with Langtry-Menter transition turbulence model including amplification factor formulation for wall roughness |
| Ti | Turbulence intensity |
| WTB | Wind turbine blades |

*Author contributions.* KV performed the CFD model development with simulations, blade element momentum calculations of eroded NREL 5MW blade, and analyzed the overall results. HM carried out the scanning of blade surfaces in the field. IB performed the blade element momentum calculations to evaluate the performance loss of the scanned blade. GS oversaw the development of the CFD transition model for 415 rough surfaces and carried out the review of the article.

*Competing interests.* The authors declare that they have no conflict of interest.

*Acknowledgements.* The first part of the research was carried out within the Dutch national project AIRTuB sponsored by the Topsector Energy Subsidy from the Ministry of Economic Affairs and Climate. The second part was funded by Nederlandse Organisatie voor Toegepast Natuurwetenschappelijk Onderzoek (TNO)'s Knowledge Innovation Program (KIP) for 2020. The authors would also like to acknowledge



the data provided by Sandia National Laboratory and offer special thanks to Dr. David Maniaci (SandiaEnergy) for providing detailed

clarification on the experimental model.



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
