# Peer review of "CFD modeling of actual eroded wind turbine blade"

_Wind Energy Science, 2022_

## Author Response (AR1)

**RC1**

Dear Beatriz,

Thanks a lot for your review and your useful comments. We have included them in the following way:

1. The different between erosion and roughness should be clarified along the manuscript.

*Roughness is a texture of the surface different from the ideal flat surface. The roughness can be both under the surface for instance caused by wear and above the surface for instance deposits and contamination. Erosion damage will have specific local effect on the surface morphology caused by wear and the shaped will vary in shape and depth under the surface.*

*Additional text has been added in the introduction, where it now reads:*

*This paper presents a two-part investigation, where the first part focuses on accurate prediction tools for simulating various roughness distributions (in the order of 0.1-0.2mm) due to light erosion or contamination and its influence on boundary layer transition. Roughness is a texture of the surface different from the ideal flat surface. The roughness can be both regress into the surface such caused by wear, and above the surface for instance by deposits and contamination. However, erosion damage has specific local effect on the surface morphology caused by wear to vary in shape and depth under the surface.*

*The erosion impact studied in the first part is assumed to be uniformly distributed at the minimal scales, where the effect of roughness is in the same order of magnitudes. Larger scales of erosion where a significant amount of material is lost cannot be assessed the same way, instead, it can be thought of as a negative imprint of roughness at larger scales, which requires the explicit modelling of the actual surfaces. The second part will focus on exactly that and demonstrates the explicit modelling high resolution scanned LE surfaces from an actual blade with LEE damage in the order of 10-20mm.*

2. In page 6 an explanation of why Cp is under predicted would be recommended.

*Assuming your comment is applicable to Fig. 8 on page 9, which is the first Cp plot from the paper. There is some underprediction of Cp for the lower 2 & 4deg AoA but seems to show negligible difference at 6 & 8deg AoA. Given the uncertainty in exactly representing the experiment, we consider the agreement as very satisfactory.*

3. Figure 5, why the stall region is not well captured? Any explanation for that?

*These results are from steady CFD simulation and it is expected of RANS simulation to fail at separated flow conditions. Time accurate simulation are required to capture stall (Wokoeck, R. et al, 2006).*

*Wokoeck, R., Grote, A., Krimmelbein, N., Ortmanns, J., Radespiel, R., & Krumbein, A. (2006). RANS Simulation and Experiments on the Stall Behaviour of a Tailplane Airfoil. In New Results in Numerical and Experimental Fluid Mechanics V (pp. 208-216). Springer, Berlin, Heidelberg.*

4. In general, in all the paper figures (starting in Figure 5)  the same text is repeated in the top part of each group of images, for instance in Figure 5 the text that describes the solver, the turbulence model, etc, is repeated, if it would be included in the legend some space will  be saved.

*We agree that it would save space, however we would like the figures to have as much information as possible if someone decided to use it as it is.*

5. In figure 12 I would recommend to plot pressure coefficients to get some more information.

*We agree with the reviewer and have now updated the publication including this Cp plot (Figure 13). Which gives further evidence of agreement with relative differences from the measurements.*

6. In figure 27 the legend repeat the word against twice, it sounds weird.

*yes indeed it sounds weird, so we rephrased it.*

7. In page 17 appears a sentence: " The novel aspect of this study is the modeling of high resolution LE surface of the WTB with LEE damage, that was captured using state-of-art optical high-resolution 3D scanner technologies. This makes this research different from numerically assumed blades with LEE damage."  Some other works have been presented previously that simulated 3D shapes using CFD, please check this reference: Experimentally validated three dimensional computational aerodynamics of wind turbine blade sections featuring leading edge erosion cavities - Campobasso - 2022 - Wind Energy - Wiley Online Library.

*The mentioned work (Campobasso,2022) indeed reports on erosion representative 3D shapes using "MATLAB script that writes a table containing the center coordinates and the diameter of each erosion cavity", which is still numerically generated geometry based on generalisation, however our work demonstrates  simulation of the actual 'pure'  3D scanned eroded surfaces on blades from the field.*

8. In page 2 this sentence is found: " The influence of surface roughness in the form of erosion or contamination is of great practical importance for many flow

applications and particular interest to the wind industry". It can create confusion since it seems that erosion and roughness is the same.

*We agree, please see answer to 1..*

**RC2**

This is an interesting article providing novel methods and results relevant to the wind energy sector. The structure and presentation of the report needs revisions. In general, the article seems to be made of two distinct 'sub-articles', each with some Introduction and Conclusions. It is recommended that the two parts be merged, which will result in a more compact and readable structure. There should be a single introduction, with a single literature survey and with a single set of Conclusions (the two sections Discussion and the section Conclusions should be merged in a single section Conclusions, for example).

*Thank you for this consideration. We have however purposely decided to write the paper as a two part study, with respective sections for each part where part 2 is a follow up of part 1, i.e. the information on modelling capabilities described in part 1 is needed to understand the analysis of part 2. We have chosen the 2 part structure because it allows the reader to focus on different part based on their independent interests*

*In order to consider your concern we have elaborated the introduction with a sort of table of content which helps the reader to understand the structure of the article and leads the reader through the article in a better way. (line 51-62)*

*We have also shifted some general information from part 1 and part 2 into the introduction.*

The technical presentation of the second part is less clear than that of the first part, and it needs to be improved. Many figures have too much information as text in the figures themselves. This information should be moved to the captions of the figures.

*WE agree that the second part is less technical but this is inevitably related to the character of the research. The first part is highly technical as it describes the numerical model and its corresponding behaviour, validation etc. but the second part mainly focuses on the measurement campaign and the validation exercise using the lessons from Part 1, hence it does not contain the same amount of technical presentation.*

The positioning of the Figures is often erratic with respect to the position of the text where these figures are cited and commented.

*We did whatever we can to improve this but we were unable to change the Copernicus Publications latex template which automates this.*

Several definitions of set-ups are missing, particularly in the second part. The literature survey is rather small and some important recent and relevant studies are omitted. The part on the scanning technology is interesting but a bit too far from the core topic of the manuscript, and probably also a bit too commercial (as presently written). Several concepts and names are defined too far down in the text with respect to the first time they are mentioned, making reading and quick understanding hard. In the following, please find more detailed comments.

Please proofread the article to remove some errors. For example, in Introduction, line 1 'occurS', and line 23 'OF particular interest'.

*We have solved this accordingly*

Abstract, line 15: The results also suggest that under fully turbulent condition the eroded LE shapes show the least amount of influence on the aerodynamic performances and results in negligible difference to AEP.This sentence should be made clearer.

*We have rephased in the abstract:*

*The results also suggest that under fully turbulent conditions, the degree of eroded LE shapes studied in this work show the minimal effect on the aerodynamic performances and results in negligible difference to AEP.*

*And we have added the following in the main body, see line 386.*

*If the boundary layer is fully turbulent there is no shift in boundary layer transition point possible which desensitises a further change of LE shapes. This results in minimal effect on the aerodynamic airfoil performance, thus resulting in negligible difference to AEP. However, this is true within the limited degree of LE shape change studied within this instigation. A very substantial change of airfoil shape will have impact on the pressure distribution and so likely to influence the airfoil performance.*

Introduction, line 21. Sentence The results also suggest that under fully turbulent condition the eroded LE shapes show the least amount of influence on the aerodynamic performances and results in negligible difference to AEP. 1 Introduction us unclear and unrelated to previous statement on the period. Please amend.

*It seems to be a sentence from the abstract which we have resolved in the above.*

Page 2, Line 34. The novel aspect of this work is the modeling of high-resolution LE surfaces of an actual blade with LEE damage, that 35 was captured in the field using state-of-art optical 3D scanning technologies. This makes this study different from numerically assumed damaged blades with standardized damage profiles on the LE. There are some more published studies using CFD to study field-recorded erosion geometry or realistic patters, e.g. references 1-3 below. The geometry used in the CFD simulations of this report were also obtained with laser scanning of eroded leading edges. This type of studies should be mentioned in the literature survey.

1— Meter Forsting et al., A spectral model generalising the surface perturbations from leading edge erosion and its application in CFD, 2022.

2— Veraart M, Deterioration in aerodynamic performance due to leading edge rain erosion, MSc thesis and Delft University of Technology, Technical University of Denmark, 2017.

3—Ortolani et al., Multi-scale Navier-Stokes analysis of geometrically resolved erosion of wind turbine blade leading edges, 2022.

*All of these article show extensive effort on generalisation of geometry based on scanned or numerically assumed geometry, which we have now included and referenced as such. The mentioned references do not report a 3D simulation of actual pure scanned geometry, as we have done on this study.*

Page 2, line 55. Some more and more gradual introduction to the objective is needed. Most readers will struggle to follow this part without a brief introduction on transition modelling for smooth and rough surfaces.

*Sure, we have included additional text gradually introducing transition modelling for smooth and rough surfaces in lines between 70-81*

Page 3, line 69. Acknowledgement of funding should go at the end of the article. Same comment applies to line before Section 3.1

*Okay, acknowledgement of funding is already mentioned at the end of the article and will be removed from the main body as to your suggestion.*

Page 3, line 79 to page end: the description of the numerical method is cluttered and needs to be clarified. Also, please use symbols for omega and other symbols and explain what they represent, citing suitable sources to reduce provision of further detail , if appropriate.

*This has now been updated and presented in Table 1*

Bottom page 3. It is said that convergence was based on all residuals dropping below 10^-7. Please also specify the residual drop, i.e. by how many orders of magnitude the residuals dropped. This is important if order of magnitude of initial residuals is not 1 for all PDEs (i.e. are these residuals normalized?).

*We included this now on line 108 -> Numerical convergence was assessed based on the absolute RMS residual for all equations < 1E-7 or a relative reduction of 5 orders of magnitudes for all simulations with a steady hysteresis on the lift and drag forces*

Page 4, line 102: far field at 90 chords. Was the choice of 90 chords made following a sensitivity study? Please provide reasons for choosing 90 chords.

*Yes indeed it follows from in-house experience/sensitivity study, and we have noted this now on line 126*

Fig. 2. Is there a special reason for modelling the flat plate geometry with nonzero thickness? Is the used thickness and the leading edge geometry the same as in the experiment? Please incorporate this information in the article.

*Yes indeed it is simulated exactly representing the experimented T3A geometry with 20mm thickness with leading edge diameter of 15mm. We have include this detail at line 117*

Fig. 5. The mesh-independent CFD analysis and the experiments are in good agreement until about 8 degrees. This is not unexpected. However, cl and cl are plotted until 20 degrees AoA, whereas the position of transition is plotted only until 8 degree AoA. If experimental data are available above AoA 8, it would be interested to include these. Also, please indicate in figure that this is position along suction side.

*As you have rightly suspected the transition location data is only available up to 8deg. "Suction side transition location" has been already indicated in the figure title.*

Page 8, line 131: perhaps SST rather than SSTLM ?

*This line correctly refers to the original SSTLM model as currently written.*

Page 8, line 134: This was accomplished by increasing the constant from 2.193 to 3.29 in the onset equation.... This sentence cannot be understood without providing further detail, e.g. which constant of which equation is affected. Also, Langel made simulations of these experimental wind tunnel tests. Did they alter the baseline SST enhanced with the transition model? If so, can the authors comment on why Langel et al, did have to alter the discussed constant to have a good agreement with measured data? Were the boundary conditions used by Langel the same as those

used by the authors? Were the grids comparable? Some discussion of this aspect would be beneficial to further improve the usefulness of the article.

*Further details on changing the onset constant has been referred to (Khayatzadeh and Nadarajah, 2014) and it modifies the original SSTLM model and not SST, as it is currently written. Langel's decision to adopt this change of constant is also noted as due to the benefits this change offer in terms of reducing the model sensitivity to free stream turbulence values.*

Page 10. Discussion on effect of variable Ar in the framework of the transition model is unclear. Too much information on the baseline transition model (without roughness) is given for granted, and for readers not knowing fairly well the baseline transition model, it may be impossible to follow the comments on the function of the variable Ar. I would suggest providing some more background information so as to make all more clear. Also, may symbols appearing in the equations are never defined, and they should be.

*The Ar variable and its transport equation is built on top of the baseline transition model – it is simply a triggering variable that prematurely forces the transition mechanism within the SSTLM model. Thus it was necessary to give much detail on the underlying transition model. On the other hand as indicated on the first sentence of section 2.3 (whole section dedicated to Ar and its implementation) the reader is informed to follow Langel et al for the detailed derivation of the model description, as it is already well document by this author. We have corrected the missing definition of symbols.*

Fig. 11. How is the TKE in the left plot defined? Is it computed at a given distance from the flat plate, is every point an average along a line normal to the flat plate ? Is it expected that this TKE is independent of the surface roughness? Please add some comments to address these questions.

*TKE is defined simply as $k = u'$ here as to comply with experimental data set from (Savill, 1993), where T3A experiments have measured the fluctuation in stream wise velocity using hot-wire probe traverse taken 30mm distance from the top of flat plate. The CFD results seem to suggest, that streamwise flow fluctuation is independent of surface roughness indeed. We have include this detail at line 121*

Page 12, line 184. Please define roughness density.

*Roughness density is percent-area coverage (area covered by the roughness element/total area). We have added this on line 215.*

Eq. 9. Did the authors of this manuscript also used this correlation between Ks and roughness height and density? If so, please state it.

*Yes, it was developed and used to generate the result on the publication. We have explicitly indicated that this eqn was used on the results to follow on line 221*

Fig. 12. Why are lift and drag curves of rough airfoil computed only up to about 8 degrees, whereas those of the clean airfoil extend to higher values of AoA? Same question applies to subsequent figures.

*The transition location data was only available up to 8deg, thus the rough airfoil simulation were only conducted at these angles of attack.*
*We have explained this on line 221*

The link to following reference appears to be broken: Maniaci, D.: https://energy.sandia.gov/programs/renewable-energy/wind-power/blade-reliability/leading-edge-erosion/, 2022.

*We have replaced this reference with the publication: Ehrmann, R. S., Wilcox, B., White, E. B., and Maniaci, D. C.: Effect of Surface Roughness on Wind Turbine Performance., Tech. rep., Sandia National Lab.(SNL-NM), Albuquerque, NM (United States), 2017.*

Page 17, line 284: sentence does not have a verb.

*We have now decluttered and rephrased the whole section 3.2 and 3.3, which resolved this.*

Page 18, lin 277: There are no known references for optical scanning systems that are already used in practice for O&M of WTBs. Please explain what does 'optimal' mean in this context. What is special about an optical system for scanning an eroded WT blade?

*We have not written the word optimal here, but there may be confusion with the word optical which we assume makes sense?*

Section 3.2 is cluttered. It contains very broad and fragmented overview of available methods for scanning blade surface.

*We have now decluttered and rephrased the whole section 3.2 and 3.3.*

Page 291, lines 291-294. This is all un clear, please provide more clear explanations.

*We have now decluttered and rephrased the whole section 3.2 and 3.3, which resolved this.*

Section 3.3 contains text repeated in Section 3.2. Please revise.

*We have now decluttered and rephrased the whole section 3.2 and 3.3, which resolved this.*

Section 3.1 should be incorporated in the Introduction at the beginning of the article. Sections 3.2 and 3.3 may be merged and moved to an Appendix.

*To our opinion this is relevant foreground information, since the scanning of a WTB with an optical scanning device is still a challenge, not standard practise (for a Appendix)*

Acronyms RANS and several others are spelt our many times in the article. These repetitions should be removed, and each acronym should be spelt out only once.

*We have removed all repetitions and all acronym are spelt out only once*

It is suggested to merge figures 27 and 28, i.e. plotting 4 curves in each subplot. This would enable a more global/general comparison.

*We have accommodated your suggestion and provided Figure 28 combining both graphs.*

Page 24, line 339. Could authors please provide spanwise length of 3D eroded blade portion modelled in CFD? Without this information, the total number of cells of the 3D simulation is not meaningful. It would also be useful to provide more detail, such as number of elements in spanwise direction and spacing, if not uniform. It is also interesting to state what the boundary conditions on the lateral walls of the domain are.

*A span length of 6\% chord was used for this study, which was modelled with approx. 150 elements. In total, approx. 700 elements were used to model both suction and pressure sides of the airfoil, where 400 elements were dedicated purely to model the LE. The clean blade section was only modeled with 3 million elements for comparison. Slip/Symmetry boundary conditions were used for the lateral walls of the domain.*

Page 24, line 346. Negligible impact on flow downstream. For which AoA does this statement hold? Please clarify.

*It's for the AoA of 10deg as represented in Fig 32, We have now added this information in line 382 too.*

Section 3.4.3, first 3 lines. Statements are unclear. Please reword/clarify/provide further detail.

*We have rephrased it in the following sense (see line 391):*

*A sensitivity study was carried out using a 2D profile extracted from the scanned 3D LE (Figure 34), and then two more artificial profiles were generated by simply performing a lateral translation of the extracted LE towards the TE. The translation distance was calculated based on the maximum lateral difference between the clean and eroded LE (Figure 35), which was found at x/c and y/c corresponding to (0,0). This*

*distance of 0.26%c  was then multiplied by 2 and 3 to generate the Eroded 2x and 3x profiles. These relative LE differences were then superimposed on a generic WTB tip airfoil NACA64-618, specifically the NREL 5MW reference turbine for the assessment of AEP.*

What is function of Fig. 32? It does not seem to provide useful information to support any statement in the manuscript.

*it supports the line 382, where It clearly shows the detail pressure resolution for the scanned simulated eroded surface, despite showing negligible difference in integrated forces.*

Results in Fig. 36 and 37. In the case of transitional analyses, it seems that the largest differences between the performance of clean and eroded airfoils are those on the drag coefficient. Assuming that Eroded2x and Eroded3x denote different erosion geometries, Figure 37 (right) shows that the transitional analysis predicts a better performance of geometry Eroded2x. This would indicate that the transition characteristics (position of transition on SS for each AoA) of geometry Eroded2x are more similar to those of the clean airfoil than those of geometry Eroded3x. This would underline the necessity of resolving erosion in AEP analyses. However, there is also the fast that small roughness is not included in these geometry-resolved analyses, and this may also alter these conclusions. Some more comments, particularly with regard to the dependence on the position of transition on AoA for the eroded airfoils, would be appreciated.

*We fail to understand this comment, the figure shows deteriorating performance with erosion. So im not sure what is meant by "better performance of geometry Eroded2x" and "the necessity of resolving erosion in AEP analyses"*

Authors show that transitional analysis predict increasing loss of aerodynamic performance moving from profile 'Measured' to Eroded2x and then Eroded3x. Can the authors please comment on what is the main cause of the performance loss increase? Is it the increasingly more jagged profile (in case the erosion profile is made more severe moving towards Eroded3x), is it the fact that thickness to chord ratio increases, something else, a combination of factors? Details on this aspect would increase the quality of this work.

*We appreciate your comment but we do not have the full answer. We think that thickness to chord ratio could play a role indeed but we also think that the jagged profile will lead to higher wall shear stresses and to an increase in momentum thickness of the boundary layer which will increase drag too*

Was a mesh sensitivity study performed for the transitional analysis of the eroded airfoils? Please add a statement on this.

*2D mesh sensitivity studies were performed for the transitional analysis of the clean airfoil (figure 5-6), while 3D mesh sensitivity study for the scanned eroded geometry was only performed with SST model.*

Lines 354-355. Some more detail on the AEP analysis, including the underlying BEM analysis, should be provided. Please provide a more clear definition of the 3 2D eroded geometries considered. Particularly, was the ragged profile scaled in passing from measured geometry to the other 2 geometries, or was a translation only performed?. The geometry of the eroded blade used in the BEM analysis should be defined more clearly. How many eroded airfoils are considered? Which is the interval of percentage radius over which each eroded airfoil is applied? (These are an example of basic data which should be reported).

*We have added more detail on line 398 as below, with a supplementary table detailing the NREL 5MW blade used for the BEM model on table 3:*

*The details of the BEM framework used for this work is documented in (Vimalakanthan, 2014). Using the same BEM model for NREL 5MW (Table 3), the tip airfoil NACA64-618's polar data was changed at different radius intervals to study the effect of erosion at different tip extents. For instance, the case of 30% eroded tip with the largest LEE was calculated by replacing all NACA64-618 sectional polar data from Table 3 with the Eroded 3x data from Figure 36.*

Page 31, line 384: Although the generated grid resolution was made to capture numerical details in the order of 0.1mm, the RANS turbulence modelling approach used 385 in this study is inherently flawed for capturing flow details at this scale of changes in geometry. I would suggest not using the word 'flawed', but rather noting there may be some uncertainty …

*Sure, we have worded it as you suggest, see line 434*

Line 391.  However, if the effect of the exposed fibrous blade material on the transitional boundary layer was included using a model like the one described in Part 1, it will further deplete the sectional L/D ratios, thus resulting in further loss in AEP.This is an interesting comment. However, is it not the case that even if erosion has not reached the structural part of the blade, the scan has a finite resolution that prevents all erosion scales to be included in erosion geometry-resolved of the type presented? Please comment on this point.

*Your comment is true and exactly to our point, due to the limitation on the scanning resolution we are not able to resolve any feature that are at smaller erosion scale. Hence the suggestion to model smaller surface distributed roughness using the model like the one described in Part 1.*

---

## Author Response (AR2)

I thank the authors for satisfactorily addressing or answering part of my comments. I would like to report the following points.

1) In my previous review I wrote: Page 2, Line 34. The novel aspect of this work is the modeling of high-resolution LE surfaces of an actual blade with LEE damage, that 35 was captured in the field using state-of-art optical 3D scanning technologies. This makes this study different from numerically assumed damaged blades with standardized damage profiles on the LE. There are some more published studies using CFD to study field-recorded erosion geometry or realistic patters, e.g. references 1-3 below. The geometry used in the CFD simulations of this report were also obtained with laser scanning of eroded leading edges. This type of studies should be mentioned in the literature survey. 1— Meter Forsting et al., A spectral model generalising the surface perturbations from leading edge erosion and its application in CFD, 2022. 2— Veraart M, Deterioration in aerodynamic performance due to leading edge rain erosion, MSc thesis and Delft University of Technology, Technical University of Denmark, 2017. 3—Ortolani et al., Multi-scale Navier-Stokes analysis of geometrically resolved erosion of wind turbine blade leading edges, 2022.

Authors replied: All of these article show extensive effort on generalisation of geometry based on scanned or numerically assumed geometry, which we have now included and referenced as such. The mentioned references do not report a 3D simulation of actual pure scanned geometry, as we have done on this study.

I would like to report to the authors that, despite what they stated above, they have not included the mentioned sources in their revised article. Perhaps the omission is accidental. If so, please amend. Please also state which of the three articles uses the 3D scan from a real turbine and which ones used scaled erosion profiles from a swirling arm rain erosion test.

In general, as I noted in my first review, the literature survey of this article is quite limited, and the authors are invited to make it a bit wider to better cover the state-of-the-art in the areas of the reported research.

*Apologies, this was in fact an accidental omission. We have now included the refences to these works and indicated the differences on the testing methods.*

2) In my previous review I wrote: Page 8, line 134: This was accomplished by increasing the constant from 2.193 to 3.29 in the onset equation…. This sentence cannot be understood without providing further detail, e.g. which constant of which equation is affected. …

Authors replied: Further details on changing the onset constant has been referred to (Khayatzadeh and Nadarajah, 2014) ...

Yes, this was already written in authors' original submission. The point is that, in this reviewer's opinion the equation containing the altered constant and possibly a couple of other ones linked to the affected equation should be reported in this article and briefly discussed. This part of the transition model is relatively complex in terms of number of equations and constants involved. Providing the few equations I am referring to would improve the effectiveness, as CFD practitioners could more quickly and with no ambiguity identify the part of the transition model affected by the constant change. It would also help CFD practitioners who do not develop this transition model but use it, to more easily identify the affected parts of the model and test the change reported by these authors and the McGill colleagues. This is why I am suggesting again to report the affected equations and constants.

*We have now included the onset equation and additional text to clarify this further (Page 9, line 165).*

3) In my previous review I wrote: Fig. 12. Why are lift and drag curves of rough airfoil computed only up to about 8 degrees, whereas those of the clean airfoil extend to higher values of AoA? Same question applies to subsequent figures.

Authors replied: The transition location data was only available up to 8deg, thus the rough airfoil simulation were only conducted at these angles of attack. We have explained this on line 221

I do not think the authors have answered my question. I did not ask about the transition location, I asked about the lift coefficient in top-left subplot of Fig. 12. Experimental data for both clean and rough cases are plotted up to AoA 16 or thereabout. CFD results of the smooth airfoil also go up to 16 deg, but rough case stops at 8 degrees. The same happens for the drag coefficients. Why was this done? Were the rough wall CFD simulations not run for AoA between 8 and 16 degrees? Could you please explain this aspect in the manuscript?

*We have now reworded the text, which reads (line 227):*

*The calibration study was conducted by using a cost function that minimizes the errors between the calculated and measured transition locations across the experimented range of angles of attacks (-4 to 6°). Transition location data was only available between -4 to 6deg within the LEES dataset, thus the rough airfoil simulations were limited within this range of angles of attack and not higher to reduce the extensive number of simulations for the calibration data. As a result, the following equation was established*

*for k$_s$ as a function of roughness height (R$_h$) and density (R$_D$ ). This calibration equation (Eqn 10) was explicitly used to generate the results presented in Figure 12 - Figure 17.*

4) In my previous review I wrote: Was a mesh sensitivity study performed for the transitional analysis of the eroded airfoils? Please add a statement on this.

Authors replied: 2D mesh sensitivity studies were performed for the transitional analysis of the clean airfoil (figure 5-6), while 3D mesh sensitivity study for the scanned eroded geometry was only performed with SST model.

OK. However, I am doubtful that for this particular problem, the results of the mesh sensitivity analysis performed on the 2D transitional problem can be 'extended' to the 3D counterpart of this problem, particularly if the 2D mesh sensitivity analysis was performed with airfoil geometries which were not slices of the 3D scan. Moreover, in the case of the 3D scan of the airfoil, there are strong geometry and flow gradients also in the third direction. In our experience, achieving grid independence for the 3D transitional case for problems of this type is not straightforward, and it requires very large HPC resources. I would recommend to mention in the manuscript what the authors write above on mesh sensitivity analyses, because the results of the 3D transitional analyses may be affected by some uncertainty they may affect the quantitative estimates of the AEP losses.

*Agreed and we have now explicitly added the following text (on line 385) to reflect on this:*

*The 2D mesh grid refinement studies were performed for the transitional analysis of the clean airfoil (Figure 5). However, due to the limited computational resources, the 3D grid refinement study for the scanned eroded geometry was only performed with SST model.*